# Single cell spatial analysis reveals inflammatory foci of immature neutrophil and CD8 T cells in COVID-19 lungs

Praveen Weeratunga[1,15], Laura Denney[1,15], Joshua A. Bull [2,15], Emmanouela Repapi[3,15], Martin Sergeant [3], Rachel Etherington [1], Chaitanya Vuppussetty[1], Gareth D. H. Turner[4], Colin Clelland[5], Jeongmin Woo[1], Amy Cross[6], Fadi Issa [6], Carlos Eduardo de Andrea[7], Ignacio Melero Bermejo [7], David Sims[3], Simon McGowan [3], Yasemin-Xiomara Zurke [8], David J. Ahern [8], Eddie C. Gamez[9], Justin Whalley [9], Duncan Richards [10], Paul Klenerman[11], Claudia Monaco [8], Irina A. Udalova [8], Tao Dong[1,12], Agne Antanaviciute [1], Graham Ogg [1,12], Julian C. Knight [9,12], Helen M. Byrne [2,13], Stephen Taylor [3,9] ✉ & Ling-Pei Ho [1,12,14] ✉

Single cell spatial interrogation of the immune-structural interactions in COVID −19 lungs is challenging, mainly because of the marked cellular infiltrate and architecturally distorted microstructure. To address this, we develop a suite of mathematical tools to search for statistically significant co-locations amongst immune and structural cells identified using 37-plex imaging mass cytometry. This unbiased method reveals a cellular map interleaved with an inflammatory network of immature neutrophils, cytotoxic CD8 T cells, megakaryocytes and monocytes co-located with regenerating alveolar progenitors and endothelium. Of note, a highly active cluster of immature neutrophils and CD8 T cells, is found spatially linked with alveolar progenitor cells, and temporally with the diffuse alveolar damage stage. These findings offer further insights into how immune cells interact in the lungs of severe COVID-19 disease. We provide our pipeline [Spatial Omics Oxford Pipeline (SpOOx)] and visual-analytical tool, Multi-Dimensional Viewer (MDV) software, as a resource for spatial analysis.

Since the first reports of COVID-19 cases in Dec 2019, the severe acute respiratory syndrome coronavirus 2 (SARS-CoV-2) has caused more than 6 million deaths worldwide[1], mainly from respiratory failure. Similarities between COVID-19 and other viral infections of the lungs like SARS and influenza have been noted, but there are specific differences which may be indicative of underlying disease mechanisms unique to COVID-19. In particular, patients with COVID-19 have excess incidence of thromboembolic disease, endothelial damage, and greater acute and long-term impact on organs other than lungs[2–5]. High-resolution immune studies in the blood have shed light on the potential mechanisms for severe COVID-19 disease, with evidence supporting myeloid cell overactivation and dysregulation, T cell exhaustion and cytokine hyperactivation[6–10]. Our recent comprehensive multi-modal study of circulating immune cells (COMBAT study)[6] and several other major studies have also concluded that a key hallmark of severity was emergency myelopoiesis[6,8,9,11,12], characterized by raised circulating immature neutrophils, cycling monocytes, and raised haematopoietic progenitors. However, it is not known how these findings in blood relate to damaged lung structural cells and other immune cells in the lungs, nor if they formed injurious immune entities.

Interrogation of the immune response in COVID −19 lungs have lagged behind studies in peripheral blood. Our understanding of the immune response in the lungs is derived mostly from several single cell and single nucleus RNA sequencing studies which have provided valuable insights on a transcriptomic level[13–18]. However, these are limited by a lack of high resolution (cell level) spatial context. Transcriptomics studies are also restricted by a lower detection rate for neutrophils as these cells possess relatively low RNA content and high levels of RNAses and other inhibitory compounds which confound their identification. Notwithstanding these limitations, studies in intact COVID-19 lung tissue are also challenging, due to the distorted lung micro-architecture and massive cellular infiltrate, making it difficult to unravel cellular connectivity and organisation. An initial evaluation of COVID-19 lung tissue using imaging mass cytometry by Rendeiro et al. concluded that there was greater spatial proximity between macrophages, stromal cells and fibroblasts in lung samples obtained later in infection but did not identify reveal further insight into the cause of severe alveolar damage[19].

In this study, we develop a bespoke mathematical package to identify statistically significant co-location between different cells, including structural cells, at the level of single cell resolution. We identify a cluster of closely apposed immature neutrophils and CD8 T cells with high immune activity, which are spatio-temporally associated with proliferating alveolar epithelium in tissue sections demonstrating diffuse alveolar damage. These findings raise the possibility of an injurious entity generated by the interaction between immature neutrophils and a specific subset of CD8 T cells in severe COVID-19 pneumonitis.

## Results

### An integrated pipeline to uncover and quantify spatial association amongst cells

Our first task was to establish a method to quantify statistically significant spatial correlations between highly-resolved immune and structural cell types in our lung tissue sections. To do this, we developed an analytical pipeline which combined an immunology-centric annotation approach with a 3-step spatial association analysis [quadrat correlation matrices (QCMs), cross-pair correlation functions (cross- PCFs) and adjacency cell network (ACN)] to provide a set of statistically rigorous spatial analytical output (Fig. 1a and Supplementary Fig. 1, and described in detail in Methods). In brief, we first used the QCM to identify cell pairs that are statistically significantly correlated in cell counts. Correlated cell pairs, of types A and B say, were then examined for co-location above random spatial association (using cross-PCF). If the cross-PCF, $g(r)$, is greater than 1, then cells of type B are observed more frequently at distance $r$ from cells of type A than would be expected under complete spatial randomness (CSR). We considered $g(r = 20)$, the value of the cross-PCF at $r = 20$, as a means of quantifying how many more cells of type B are observed at distance 20 μm from an anchor cell of type A than under CSR. We then examined whether the co-locating cell pairs were physically in contact with each other using a spatially embedded 'adjacency cell network' (ACN). Using the ACN, we computed the proportion of cells of type A that were in contact with at least one cell of type B, (full description found in Methods).

These work packages were integrated computationally into a workflow of Python and R based command line tools which may be run individually or as an automated pipeline (Spatial Omics Oxford pipeline; SpOOx) (Fig. 1a). The pipeline is supported by a visualisation platform [Multi-Dimensional Viewer (MDV)] (Video V1). Both are available as an open access online resource (see Methods for link). To differentiate this lung-based from our blood-based study (COMBAT)[6], we have called this the COSMIC (COVID-19 Lung Single Cell Mass Cytometry Imaging Consortium) study.

### Histopathology states of inflammation, damage and repair are found in lung sections at point of death

We started by examining formalin-fixed paraffin-embedded (FFPE) lung sections from a cohort of patients who died from PCR-positive COVID-19 pneumonitis from one hospital (University of Navarra, Spain) ($n = 12$). Samples were obtained at the point of death and fixed immediately, markedly reducing post-mortem tissue deterioration[20]. All samples were collected during the first wave of the pandemic in 2020, before vaccination and repeat infection with SARS-CoV-2. Healthy lung sections from patients undergoing lobectomy for early, isolated lung cancer (HC) lungs ($n = 2$) were used as comparators (Demographics in Supplementary Table 1); obtained from the Oxford Radcliffe Biobank (Oxford University Hospitals NHS Foundation Trust, UK).

Six of 12 patients were mechanically ventilated (range 6–23 days). All but three were receiving corticosteroids at the point of death (Supplementary Table 1). In all patients, thoracic CT scans closest to the day of death demonstrated typical and extensive COVID-19 pneumonitis comprising ground glass changes and consolidation (Supplementary Table 1). Five of 12 lung sections showed evidence of both PCR and immunostaining for SARS-CoV2 Nucleocapsid protein; 3 were PCR+ but protein negative (Supplementary Table 2). Four sequential lung sections (6 μm thick) were used for haematoxylin and eosin (H&E), 37-plex panel staining (35 metal-tagged antibodies and 2 DNA chelators), and selected immunofluorescence validation sequentially.

Initial histopathology analysis (independently performed by two senior pathologists with expertise in lung and infectious disease, and a senior respiratory clinician) revealed a highly distorted lung architecture with extensive cellular infiltrate in all samples, changes previously observed in post mortem studies of COVID-19 lung sections[21–26]. However, all our sections can be categorised into three formal histopathology classifications of predominantly alveolitis (ALV), diffuse alveolar damage (DAD) or organising pneumonia (OP) ($n = 4$ patients in each category)[27,28] (Fig. 1b and Supplementary Fig. 2). ALV was characterised by thickened alveolar epithelial wall and septae with immune cell infiltrate and congestion of alveolar walls; DAD, by widespread alveolar epithelial lining injury accompanied by hyaline membrane, regenerating/proliferating Type II alveolar epithelium and interstitial oedema, while OP depicts a repair state typified by presence of fibroblasts, proliferation of alveolar epithelium and collagen presence around bronchial epithelium[28]. In keeping with this, patients with dominant OP histopathology showed a trend of being sampled furthest away from their first symptoms (Fig. 1c), had longer periods of stay in hospital and were mechanically ventilated for longer (Supplementary Table 1 and Supplementary Fig. 3) (no statistical difference observed). All 5 patients with evidence of dual SARS-CoV-2 N protein and PCR expression had lung sections that showed DAD. No sections with OP were positive for SARS-CoV-2 protein staining (Supplementary Table 2; Supplementary Fig. 4). There were no associations between histopathology states and clinical features (age, drugs used, co-morbidities, or C-reactive protein (CRP) nearest the point of death) (Supplementary Table 1, Fig. 1D).

These results provide a histopathology-based temporally progressive states for further analysis. 2–3 regions of interest (ROIs) per patient (total of 4 mm2 area per patient), selected as representative areas for the dominant histopathology state, were drawn for ablation.

### Identification of immature neutrophils−CD8 T cell clusters with high immune activity

After preliminary staining with an initial panel (Supplementary Table 3, Supplementary Fig. 5) on a 'sentinel' cohort, we designed a final panel which incorporated the most abundant structural and immune cell types (Supplementary Fig. 7A). Single cell segmentation performed

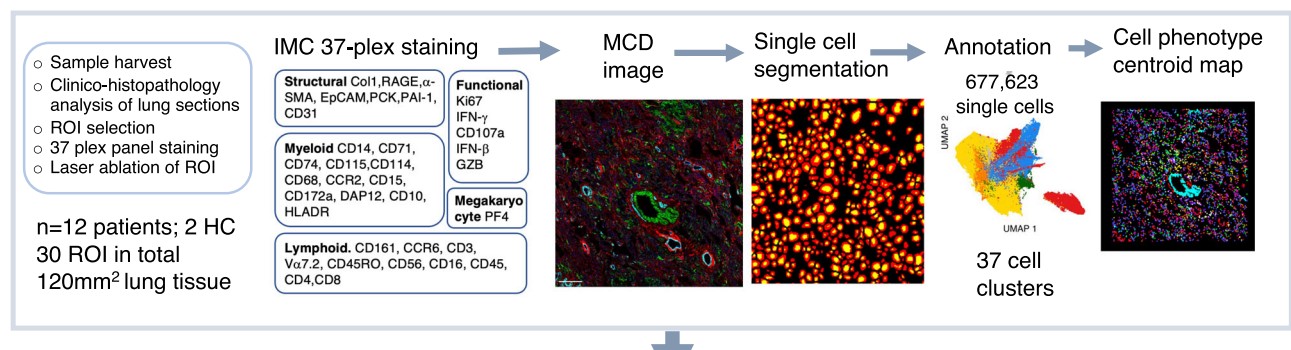

**Fig. 1 | Spatial analysis pipeline and histopathology categorisation of samples.**
**a** Overview of the workflow and SpOOx pipeline. The steps of the analysis are presented in Supplementary Fig. 1 in more detail. $n = 677,623$ single cells refer to segmented cells, without filtering for cells with no antibody staining, and 'undefined' clusters. IMC – imaging mass cytometry. **b** H&E section from COVID-19 tissue section showing formal histopathology features of alveolitis (ALV), diffuse alveolar damage (DAD) and organizing pneumonia with their corresponding MCD file image showing staining for 5 of 35 antibodies (α-SMA, EpCAM, PanCK, Col 1a and CD31). 'a'–'c' in figure refer to characteristic features of ALV, DAD and OP. 'a' - thickened alveolar epithelial wall and septae with immune cell infiltrate and congestion of alveolar walls 'b' widespread presence of hyaline membrane, and regenerating/ proliferating Type II alveolar epithelium and 'c' -fibroblasts and collagen presence around bronchial epithelium. See also Supplementary Fig. 2. Representative H&E

and MCD images is for ROI from $n = 10$ ROIs for ALV, $n = 8$ ROIs (DAD), $n = 8$ ROIs (OP); $n = 12$ patients. H&E staining performed once per tissue section. 37-plex staining was performed once for each lung sample. **c** Point when samples were obtained from the first day of symptoms and corresponding histopathology states in lung sections. Mean and S.D. shown, $p$ value calculated using one-way ANOVA test with Tukey's multiple comparison test; normality tested with d'Agostino & Pearson test. $n = 4$ patients in each histopathology group (ALV, DAD and OP). **d** C-reactive protein (CRP) levels closest to the point of sampling and corresponding histopathology state in lung sections. Median and IQR shown, $p$ value calculated using Kruskal-Wallis test with Dunn's multiple comparison test. $n = 4$ patients in each histopathology group (ALV, DAD and OP). Source data are provided in the Source Data File.

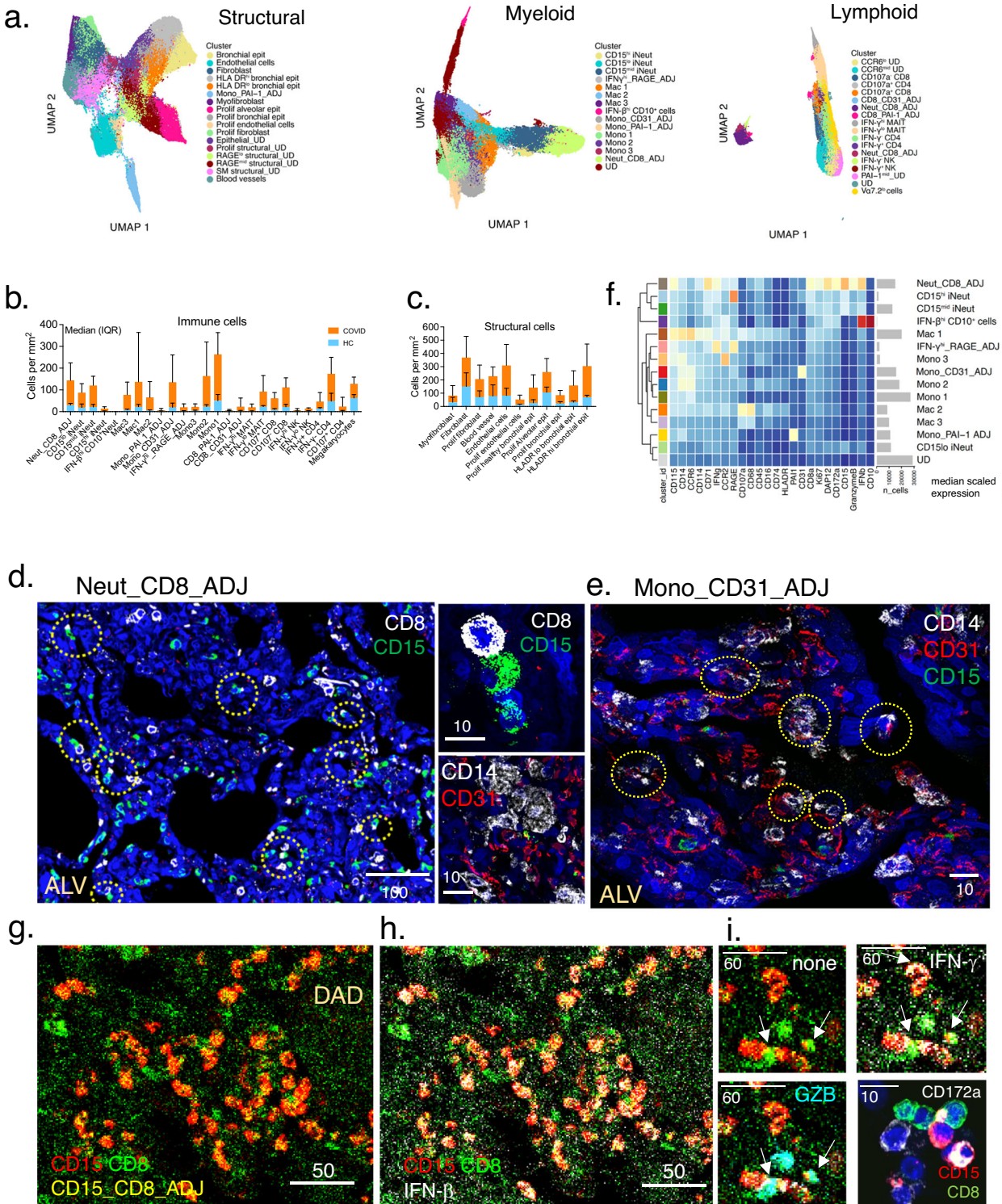

using Mesmer library from the DeepCell algorithm[29] resulted in 677,623 single cells from all ROIs, [ALV ($n = 10$ ROIs), DAD ($n = 8$), OP ($n = 8$) and HC ($n = 4$)] (Fig. 2a, Supplementary Fig. 7 and 12). Cell clusters were derived using Phenograph and annotation was performed using a combination of expression heat map analyses and expression density plots (Supplementary Fig. 7). Final annotation was refined with Pseudotime analysis of selected groups of cell clusters, examination of distributions of the cell clusters in all samples, and cross-checking with H&E and MiniCAD Design (MCD) images (generated by the Hyperion imaging system) against known structural cell location and cell morphology (Supplementary Fig. 7). This produced a

final list of 37 identifiable cell clusters (26 immune cell types and 11 structural) (Fig. 2b, c). Expanded description of the annotated cells is provided in Supplementary Table 4. For clarity of terminology, once the cell clusters were annotated, they were termed 'cell types' or 'cells' unless there were more than two cell types in the annotation.

Compared with healthy lungs (without dividing into different histopathology states), monocytes were the most abundant immune cells (Fig. 2b). Amongst the annotated cell types, five were found to co-express defining markers of different immune or structural cells, reflecting closely located or apposed cell groups (Neutrophil and CD8 T cell, Monocyte and CD31+ cells,

**Fig. 2 | High definition immunophenotyping of lung cells and identification of tissue structure. a** UMAP representation of myeloid, lymphocyte and structural cell 'mega clusters' from all regions of interest (ROI) (k = 30) (COVID-19 and HC). See also Supplementary Fig. 7 for extended analysis steps. HC – healthy control. **b, c** Number of cells per mm² of lung tissue sections in all COVID-19 samples (n = 12 patients, 30 ROIs' in total) compared to healthy control (HC) samples (n = 2 individuals, 4 ROIs in total). Median shown, error bars are IQR. n = 524,552 cells in total for COVID-19 samples, n = 30,053 cells for HC. Statistical analysis performed after samples grouped into histopathology states (see Fig. 3d). Source data are provided in the Source Data File. **d, e** Immunofluorescence (IF) staining validation for Neut_CD8_ADJ cell cluster and Mono_CD31_ADJ cell clusters. Small panels are high magnification confocal images showing CD8 and CD15 staining (top small panel), and CD14 and CD31 staining (bottom small panel) on adjacent cells. Broken yellow circles show CD8 T cell (white- CD8) – neutrophil (green-CD15) couplets throughout lungs (**d**); and CD14-staining cells next to CD31-expressing cells (endothelium) in lung tissue (**e**). See also Supplementary Fig. 6 for negative controls. IF images shown are representative of lung sections from n = 3 patients; staining experiment performed once per lung sections. **f** Heatmap of median scaled intensity for each marker for all cell clusters in the 'Myeloid' mega cluster. 'n_cells' - average number of cells in all COVID-19 ROIs. Total cells – 171, 777. UD – undefined cluster. **g, h** Exemplar MCD image from 37-plex imaging mass cytometry (IMC) staining of a DAD ROI showing expression of CD8 T cells (CD8 -green), neutrophils (CD15-red) and CD8_CD15_ADJ cell clusters (green and red co- expression, making yellow). Image is one of n = 26 ROIs, some of which do not have the CD8_CD15_ADJ cell clusters—see Fig. 3b for number of ROIs showing presence of this cell cluster in all ROIs (n = 26 COVID-19; n = 4 HC). **h** Same MCD images as (G) but with IFN-β channel 'open' (white) showing IFN-β expression on Neut_CD8_ADJ (yellow). **i** Higher magnification of a set of 3 MCD panels - 'none' - Neut_CD8_ADJ (yellow) only (arrows); 'IFN-γ' – with 'IFN-γ' (white) channel opened on MCD viewer showing expression on Neut_CD8_ADJ (yellow) (arrows) and some CD8 (green); 'GZB' - with 'GZB' (cyan) channel opened and showing expression on Neut_CD8_ADJ (yellow) (arrows). CD172a panel shows confocal immunofluorescence staining (white) on CD15 and CD8 adjacent to each other. IF images shown are representative of lung sections from n = 3 patients; staining experiment performed once per lung sections. ALV – alveolitis, DAD – diffuse alveolar damage, OP -organising pneumonia. MCD images from all 26 ROIs (n = 10 ALV, n = 8 DAD and n = 8 OP) were analysed and median expression intensity for all ROIs shown in (**f**) and Supplementary Fig. S7. All scale bars in μm.

Monocyte- PAI-1⁺ cells, CD8 T cells – PAI-1⁺ cells and IFN-γ^hi cells and RAGE⁺ cells). We labelled these 'adjacent' (ADJ) cell types. Immunofluorescence staining confirmed presence of two different adjacent cells for the Neut_CD8_ADJ (immature neutrophil and CD8 T cells) (Fig. 2d, and Supplementary Fig. 7L). Mono_CD31_ADJ comprised both monocytes that were found adjacent to endothelial cells and CD31-expressing monocytes (Fig. 2e).

Of note, the Neut_CD8_ADJ cells contained the most immature neutrophil cell type (CD71^hi neutrophils) coupled with CD45RO⁺ CD107a⁻CD8 T cells (Fig. 2f). The cluster also had the highest expression of Granzyme B (GZB), CD172a (SIRPA), IFN-β and IFN-γ (Fig. 2f, g-I). Within the cluster, the GZB expression was found on the CD8 T cells, indicating these as cytotoxic CD8 T cells (Fig. 2I). The monocyte subset in the Mono_CD31_ADJ cluster was the least differentiated (to macrophage) monocyte subset; similar to the Mono_1 cell type (Fig. 2d, Supplementary Fig. 7L).

## High innate immune cell numbers found in all histopathology states

We next sought to understand how immune cell abundance changed as the overall histology progresses from injury to repair. Firstly, we observed that changes in numbers of structural and relevant immune cells supported the temporal progression of histopathology states from inflammation to damage and subsequent repair (Fig. 3a). There was a progressive increase in numbers of all subsets of macrophages, fibroblasts, proliferating fibroblasts and myofibroblasts from ALV to DAD to OP, consistent with transition from tissue injury to repair. Endothelial and proliferating endothelial cells, proliferating bronchial epithelium and bronchial epithelium also increased progressively. Changes in abundance of macrophages over the three histopathology states reflected accumulation of macrophages as monocytes differentiate into macrophages with progression of disease.

Across the three histopathology states, we found high numbers of classical monocytes, immature neutrophils, and some subsets of MAIT, CD4 and CD8 T cells. The most significant progressive increase in numbers across the histopathological states (compared to healthy lungs) was observed for CD8 T cell subsets and CD8 containing ADJ cell clusters, and CD107a⁺ CD4 T cells (Fig. 3b–d) (see Supplementary Table 4 for expanded phenotype description of immune cell types). Neut_CD8_ADJ cluster was increased from the earliest histopathology state and remained high in all states. Apart from IFN-γ^lo MAIT cells, there were only small numbers of MAIT and NK cell subsets (Fig. 3c). Cycling (Ki67⁺) monocytes were not found in the lungs.

Overall, innate cell numbers in the infiltrate did not decline despite disease progression and were accompanied by increasing numbers of CD8 T cells [even though viral protein was absent in OP (repair) samples]. Some immune correlates of severity in the blood observed in other studies (cycling monocytes, NK cells, and activated MAIT cells) were not found in significant numbers in the lungs.

## Distinct spatial organisation found between immune and structural cells

To determine if the cells showed spatial association and organisation amongst themselves, we employed spatial statistical algorithms (Fig. 1a and Supplementary Fig. 1) to (a) understand which immune cells were found co-located with injured structural cells, (b) explore how immune cells organise amongst themselves, and (c) for the immune cells implicated in severe disease from COMBAT (monocyte, megakaryocyte, MAIT, CD4, CD8 and neutrophil subsets), determine where these were co-located or physically interacting with other immune or structural cells

In total, we found 3888 non-replicate pairs of cell types (mono1:mono1 and CD15^hi iNeut:CD15^hi iNeut were examples of pairs of identical cell types, and filtered out) in the three histopathological states (ALV, DAD and OP). Using our three-step spatial analysis, 357 pairs of cell types were identified as statistically correlated in the QCM analysis (FDR < 0.05). These cell pairs were submitted for cross-PCF analysis, with one cell type in the pair defined as the 'anchor cell' -the cell against which statistically significant connections were quantified. By pre-analysis consensus, pairs of cell types with borderline statistical significance i.e. FDR values between 0.05 and 0.10 were also submitted to prevent loss of biologically relevant data from hard mathematical cut-off. The resulting co-located cell pairs were divided into 'structure:immune' pairs (structural cells were designated the 'anchor' cell type) (n = 33) and 'immune: immune' pairs (one cell type in one of the duplicate pairs was designated the anchor cell type; e.g. for the CD107a⁺CD8:mono1 pair and mono1: CD107a⁺CD8 pair; CD107⁺CD8 in the former pair was made the anchor cell, and the latter pair was excluded) (n =117) (Fig. 4a). 'Structure' cells of interest were the key structural cells that were known to be inflamed or damaged in COVID-19 pneumonitis – endothelium ('Endothelium' and 'proliferating endothelium') and larger blood vessels ('blood vessels'), alveolar epithelial cells ('proliferating alveolar epithelium') and bronchial epithelial cells ('HLA DR^hi bronchial epithelium' 'HLA DR^lo bronchial epithelium' and 'bronchial epithelium'). We were particularly interested in 'proliferating alveolar epithelium' as their markers and location in the

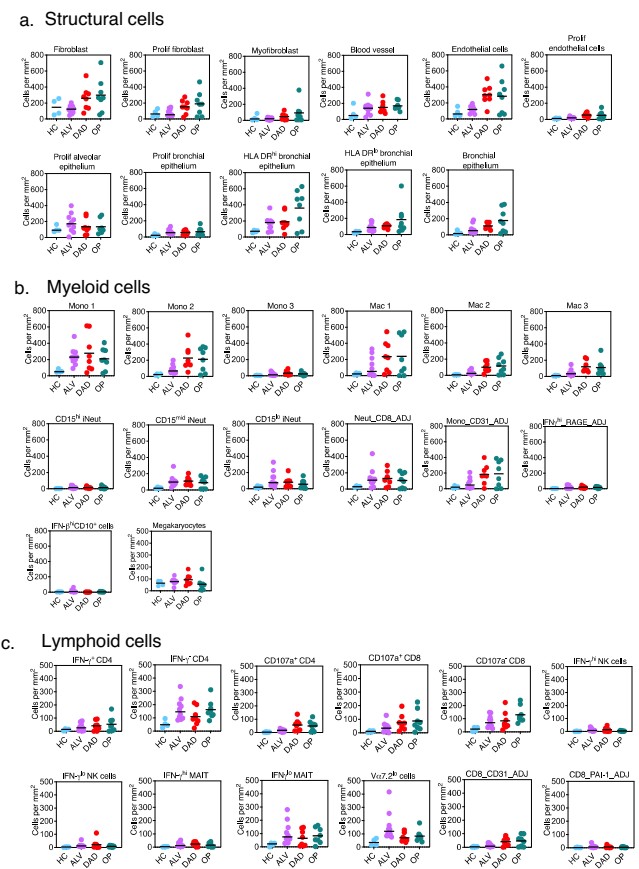

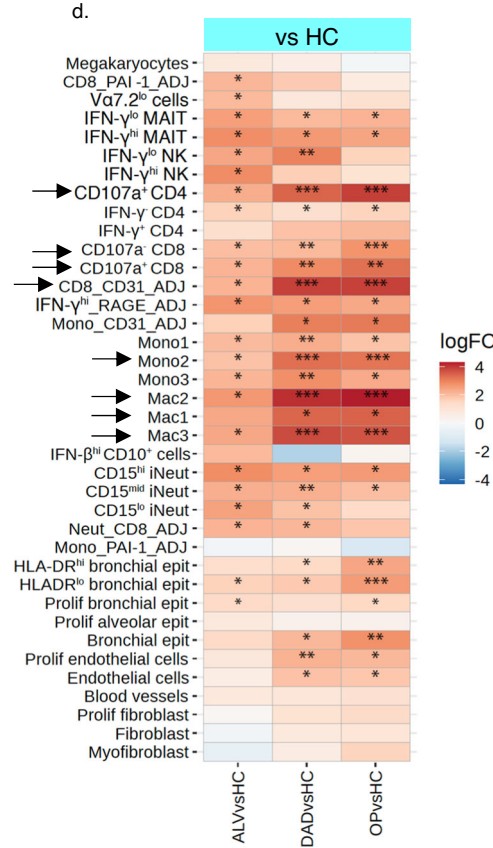

**Fig. 3 | Quantification of immune and structural cells in COVID-19 lungs.**
**a**−**c** Cell abundance plots for immune cells (myeloid and lymphoid cells) and structural cells in lung tissue, adjusted for surface area in COVID lungs categorised into those with histopathology states of alveolitis(ALV) (n = 4 patients, 10 ROIs), diffuse alveolar damage (DAD) (n = 4 patients, 8 ROIs) and organising pneumonia (OP) (n = 4 patients, 8 ROIs), compared to healthy control (n = 2 individuals, 4 ROIs). Line in figure represents median. See Supplementary Table 4 for extended phenotypic description for all cell types and clusters. Source data are provided in the

Source Data File. **d** Heatmap of fold change (FC) difference in abundance of cell types for COVID-19 samples (ALV, DAD and OP) vs healthy controls (HC) depicted in (**a**). Asterisks show those with significant differences - adjusted p values are *p < 0.05 **p < 0.01 ***p < 0.001, calculated using code from the diffcyt R package (version 1.8.8) with the option testDA_edgeR; two-sided analysis employed, and multiple comparisons adjusted using Benjamini-Hochberg method. Arrow refers to immune cells that showed progressive increase in abundance with progression histopathology states from ALV to OP.

lung sections suggest they were likely the type II alveolar epithelial cells, the purported progenitors (or stem cells) of alveolar epithelium (Supplementary Fig. 7J and K). The ability of these cells to differentiate to type 1 alveolar epithelium is critical to normal repair and alveolar regeneration after viral induced damage[30–32].

Amongst the immune cells, the strongest co-location, depicted by g(r = 20) > 2 [i.e. >2 times more cells of type B observed at 20 µm from cells of type A (anchor cell) than expected under complete spatial randomness], was observed for pairs of immune cell types that belonged to the same immune phenotype, e.g. Mac1 and Mac2 (macrophages), and the CD4 and CD8 T cell types (Fig. 4b). This was expected biologically and provided a degree of validation for the mathematical analysis. For example, close association between helper CD4 T cells (IFN-γ + CD4 T cells) and cytotoxic CD8 T cells (CD107a + CD8 T cells) is expected as the former plays critical roles in aiding the latter's anti-viral activities. However it is notable that this close physical relationship persists in OP, despite lack of viral protein at this stage of the disease (Supplementary Fig. 9).

These results signify presence of specific spatial organisation for several immune and structural cells despite appearance of disorder in tissue. The strongest co-location between all cells was found between CD4 and CD8 T cell subsets, particularly active effector memory CD4 T cells (IFN-γ+ CD4 T cells) and cytotoxic CD8 T cells (CD107a+ CD8 T cells), which did not lessen with progression to repair, and despite absence of viral proteins.

## Immature neutrophil-CD8 T clusters are co-located with proliferating alveolar epithelium in regions with maximal alveolar damage

For the significantly co-located pairs of cells, we next questioned which immune cells were found co-located with injured structural cells. To provide a composite view of the multiple outputs from our spatial analysis, we generated a 'spatial connectivity plot' to show all cell types that were statistically co-located with a designated 'anchor cell type'. Each spatial connectivity plot displayed the strength of co-location [g(r = 20)] and the average count for the immune cell types in the histopathology state (Fig. 5a, b). The proportions of co-locating cell types which were in direct contact with the anchor cell type were calculated with the ACN analysis (see Methods) and shown in the accompanying histograms (Fig. 5c, d).

Our main structural cell types of interest were the Ki67+ proliferating alveolar epithelial cell and endothelial cells. Designating proliferating alveolar epithelium as the anchor cell, we found CD15hi iNeut, Mono_CD31_ADJ and Neut_CD8_ADJ to be significantly co-located with proliferating alveolar epithelium in DAD (Fig. 5a, b) [g(r = 20) > 1]. Of these cells, proliferating alveolar epithelium was most in contact with Mono_CD31_ADJ (average of 17.6% of proliferating alveolar epithelial cells in DAD) and Neut_CD8_ADJ (8.9% of proliferating alveolar epithelial cells in DAD) (Fig. 5c). There was also a small number of IFN-γhi_RAGE_ADJ cells found co-located with proliferating alveolar

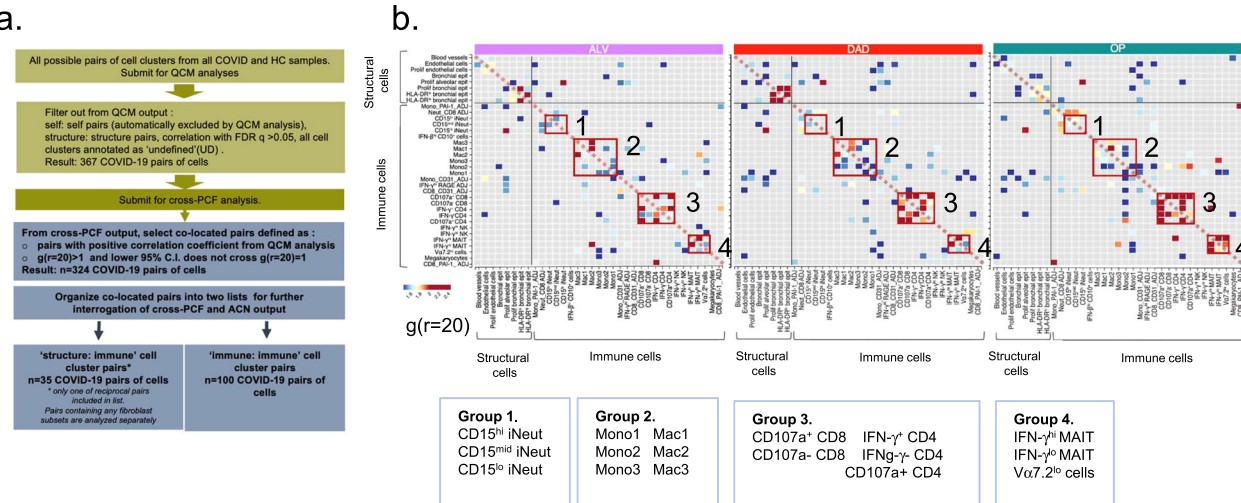

**Fig. 4 | Spatial analysis of immune and structural cells in COVID-19 lungs.**
**a** Schematic representation of the sequential spatial analysis of cellular co-location, starting with quadrat correlation matrix (QCM), then cross pair correlation function (cross-PCF) analysis, interrogation of cross-PCF output and organization according to main questions. QCM output is provided in Supplementary Fig. 8. **b.** g(r = 20) heatmaps showing statistically significant correlated pairs of cells derived from

QCM and cross-PCF analysis (see Methods for full description). n = 479,349 single cells from n = 12 COVID patients' lung sections (n = 26 ROIs); in total, n = 144,937 cells in ALV, n = 146333 in DAD and n = 163,506 in OP. Red boxes indicate groups of cell subsets from the same immune phenotype—neutrophils (Group 1), monocytes and macrophages (Group 2), CD3 T cells (Group 3) and MAIT cells (Group 4).

---

epithelium in all histopathology states, which could be resident alveolar macrophages found along alveolar epithelium.

For endothelial cells (which encompassed the smaller capillaries and the larger blood vessels in the lungs), the co-locating cell types with highest g(r = 20) in DAD were Mono_CD31_ADJ (2.1) and Mono_PAI-1_ADJ (1.6) clusters (Fig. 5b, f). ACN analysis showed more of the endothelial cells were

physically in contact with the Mono_PAI-1_ADJ cluster (21.2%) than Mono_CD31_ADJ (16.5%) in DAD (Fig. 5d). Mono_CD31_ADJ cells showed significant spatial association with endothelial cells across all histopathology states.

Next, we designed a 'radial connectivity map' to provide an overview of all immune cells that were significantly co-located with all structural cells and their corresponding histopathology states (Fig. 5g). Using this map, and focusing on proliferating alveolar epithelium and endothelial cells, we observed that while the monocytes (and their subsets and ADJ clusters) were mainly found co-located with both alveolar epithelium and endothelial cells, immature neutrophils were found predominantly with proliferating alveolar epithelium. We also observed that besides proliferating alveolar epithelium, the Neut_CD8_ADJ cluster was not found with any other structural cell types.

Finally, we developed a topographical correlation map (TCM) (Methods, Supplementary Fig. 15) to visualise how the spatial correlation between Neut_CD8_ADJ and proliferating alveolar epithelium changed across an ROI (Fig. 5h). We observed marked heterogeneity in the strength of correlation for this pair of cell types across the tissue.

One other cell type of interest was the megakaryocyte. These CD34⁻ platelet precursors, a product of emergency myelopoiesis, were the most abundant immune correlate in the blood in the COMBAT study[6]. Examining their spatial connections with our two structural cells of interest, we observed that megakaryocytes were associated with endothelium in DAD (Fig. 5g).

Drawing these data together, our spatial analysis identified Neut_CD8_ADJ and Mono_CD31_ADJ clusters as key spatial correlates with proliferating alveolar epithelium in DAD. A visual exemplar of this co-location of Neut_CD8_ADJ and alveolar epithelium is shown in Fig. 5I. Mono_CD31_ADJ and Mono_PAI-1_ADJ were the strongest spatial correlates with endothelial cells, the

former was the case across all states. No immature neutrophils (alone or in an ADJ cluster with CD8 T cells) were found with endothelial cells in any histopathology states. It is noteworthy that there was no significant co-location between any immune cells and the larger blood vessels; nor between CD107a⁺ CD8 T cells and IFN-γ⁺ CD4 T cells with proliferating alveolar epithelium or endothelial cells despite relatively high abundance in the tissue. In addition, despite a correlation with disease severity in the blood, NK and MAIT cells did not co-locate with any structural cells. Further, even though macrophage subsets were the most abundant cells in lungs, there was also no statistically significant co-location between these cells and damaged structural cells.

All data, the spatial connectivity plot, radial connectivity map, and topographical correlation map functions are available as open resources on MDV (https://mdv.molbiol.ox.ac.uk/, Supplementary Fig. 10, "Methods").

## Immature neutrophils have a spatial predilection for CD8 T cells

We next examined how immune cells connected to other immune cells by interrogating the 91 pairs of immune cells with g(r = 20) > 1 across the three histopathology states (Fig. 6a–c).

We observed that as single entities (as opposed to those found within ADJ clusters), immature neutrophils only co-located with CD8 T cells or CD8-ADJ clusters (Fig. 6a), regardless of histopathology state. However, immature neutrophils within the Neut_CD8_ADJ cluster, co-localised with Mono_CD31_ADJ clusters in DAD and other monocyte subsets in OP (Fig. 6a, d).

Therefore, in DAD, proliferating alveolar epithelium not only co-located with Neut_CD8_ADJ, but also with a further network of co-locating immune cell types linked to the Neut-CD8_ADJ cluster, forming a super network of Neut_CD8_ADJ and Mono_CD31_ADJ clusters around the proliferating alveolar epithelial cells. This can be seen in the ACN analysis (Fig. 6f) and an MCD image view of the cells in the tissue (Fig. 6g).

In contrast to neutrophils, there was a less restricted repertoire of co-locating cell partners for monocytes. Monocyte subsets and ADJ clusters were found co-located with NK, MAIT, CD4 and CD8 T cell subsets (Fig. 6b, c). Notably, megakaryocytes were found uniquely associated with Mono_CD31_ADJ in DAD (Fig. 6e).

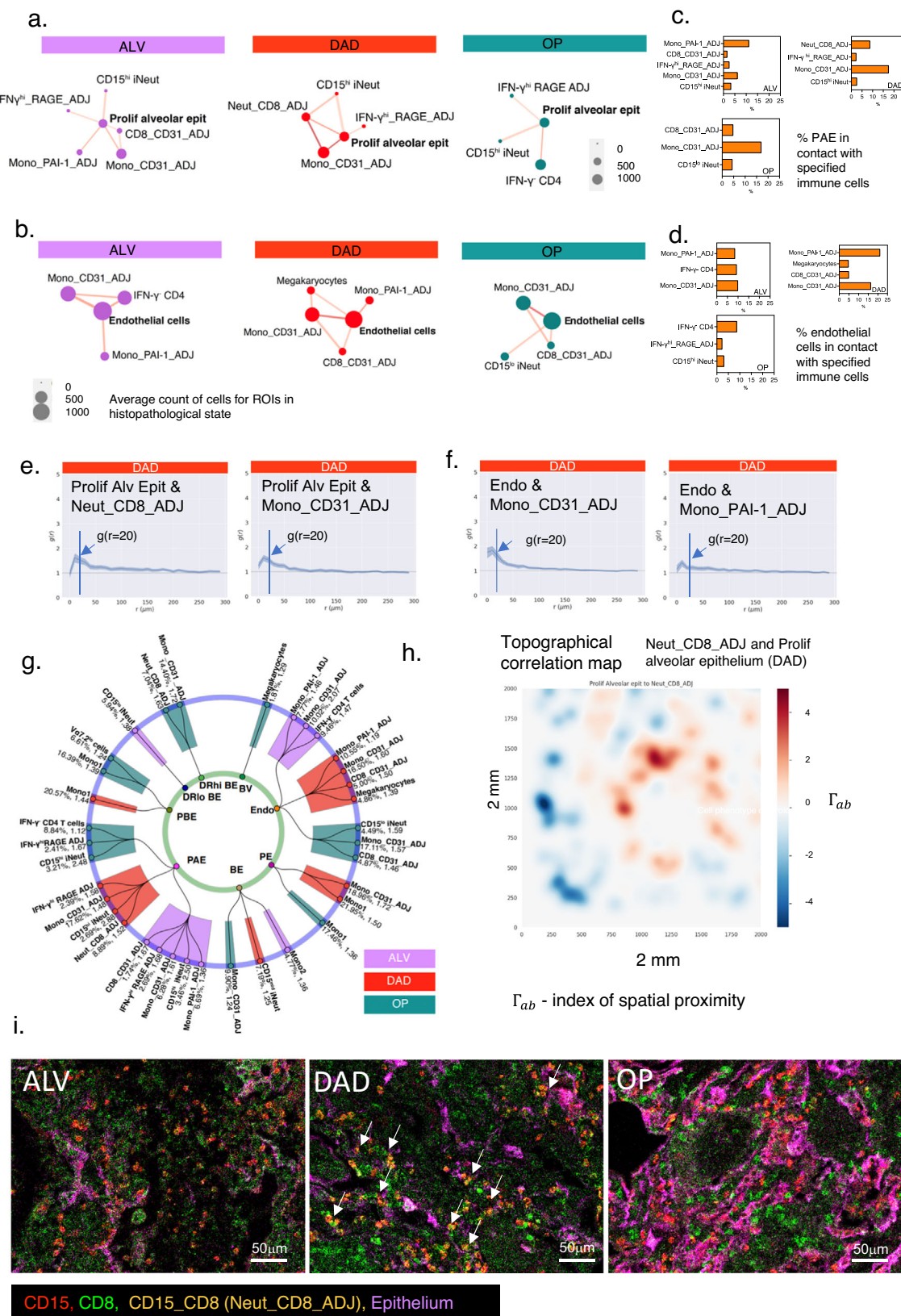

Γ_{ab} - index of spatial proximity

Our analyses showed that there were distinct organisations amongst immune cells in COVID-19 lungs, with specific predilection of immature neutrophil for CD8 T cells, and upon connection (as the neutrophil_CD8_ADJ cluster), a further connection with Mono_CD31_ADJ cluster was formed, resulting in a network of Neut_CD8_ADJ and Mono_CD31_ADJ, linked to proliferating alveolar epithelium in

diffuse alveolar damage. These were then linked to megakaryocytes via the latter cell type's connection with Mono_CD31_ADJ cluster in DAD. Thus, a spatial network of immature neutrophils, CD8 T cells, classical monocytes and megakaryocyte form a connected web of cells juxtaposed against proliferating alveolar epithelial cells and alveolar capillaries in DAD.

**Fig. 5 | Spatial organization of immune cells around structural cells in COVID-19 lungs. a**, **b** Spatial connectivity plots for proliferating alveolar epithelium, showing immune cells that are significantly co-located to proliferating alveolar epithelium (designated 'anchor cell') in the three histopathology states. The size of the nodes (filled-in circle) represents mean cell counts (abundance) for the specified cell cluster for all the ROIs in the histopathology state (scale shown in grey), and colour of nodes relate to histopathology state. Connecting lines indicate a statistically significant co-location between the two cell types derived from QCM and cross-PCF analyses. The thickness of the lines relates to the g(r = 20) value relative to each pair in the plot – the thicker the line, the higher the g(r = 20) and therefore greater strength of co-location between the immune cell type and anchor cell. n = 479,349 single cells from n = 12 COVID patients' lung sections (n = 10 ROIs for ALV; n = 8 DAD; n = 8 OP); n = 144,937 cells in ALV, n = 146333 in DAD and n = 163,506 in OP. Histogram shows % of two anchor cells – proliferating alveolar epithelial (PAE) cells (**c**) and endothelial cells (**d**) that are in contact with specified immune cell type. Source data are provided in the Source Data File. **e**, **f** Cross-PCF profiles for the two most abundant co-located structure:immune cell pairs in DAD. Curves show the change in g(r) along the radius(r) from anchor cells [proliferating alveolar epithelium (prolif alv epit) and endothelial cells (endo)] for Neut-CD8_ADJ cell clusters and Mono_CD31_ADJ cell clusters respectively. Blue coloured area

around curve is the 95% confidence interval for n = 8 ROIs with DAD. **g**. Radial connectivity map depicting all statistically significant pairs of structure:immune cells in all histopathology states; anchor cells (structural cells) are in smaller, inner circle. n = 479,349 single cells from n = 12 COVID patients' lung sections (n = 10 ROIs for ALV; n = 8 DAD; n = 8 OP). 'DRhi BE' – HLADR^hi bronchial epithelium; 'DRlo BE' – HLA DR^lo bronchial epithelium; "Endo'- endothelial cells; 'PAE'- 'proliferating alveolar epithelium', 'PBE' – 'proliferating bronchial epithelium'; 'PE' – 'proliferating endothelium" 'BV" –'blood vessels'. Numerical values indicate g(r = 20) for that pair in that state (coloured bar), and % indicates proportion of anchor cells that are co-located with the specified immune cells. **h** Topographical correlation map showing distribution of the co-located Neut_CD8_ADJ cluster and proliferating alveolar epithelial cell pair (left panel) in an exemplar tissue (an ROI with DAD). Cells of type A (e.g. Neut_CD8_ADJ) are positively ($\Gamma_{ab} \gg 0$) or negatively ($\Gamma_{ab} \ll 0$) associated with cells of type B (e.g. Proliferating alveolar epithelium) (see Methods). **i**. MCD images showing Neut_CD8_ADJ clusters amidst single CD8⁺ T cells, CD15⁺ immature neutrophils and epithelial markers (EpCAM and PanCK). Couplets of CD8⁺ and CD15⁺ cells - Neut_CD8_ADJ clusters (red and green merging to form yellow cells) (arrows) are most clearly visible in DAD. Exemplar section is shown from analyses of n = 10 ALV ROIs, n = 8 DAD ROIs and n = 8 OP ROIs (n = 12 patients). Sections were stained once with 37 plex panel.

## Projection of the circulating source of lung CD8 T cells, monocytes and immature neutrophils

Finally, we returned to our COMBAT data[6] to explore if we can identify the circulating source of the monocytes, CD8 T cells and neutrophils found in the lungs. Using *scmap*, a method which enables label projection by calculating the similarity between cells profiled by two separate studies[33], we examined the phenotypic similarity between monocytes and CD8 T cells in the lungs [this study (COSMIC)] and blood (COMBAT study). For COMBAT, we used the CYTOF dataset from neutrophil-depleted whole blood (Supplementary Fig. 3 in COMBAT)[6].

Both lung CD107a⁻ CD8 and CD107a⁺ CD8 matched to blood 'GZB^neg CD8 T cells' in COMBAT (Fig. 7a). Lung IFN-γ + CD4 T cells matched to COMBAT's 'activated CD4 T cells' subset (which contained CD27⁻ and CD27⁺ CD4 T cells). All monocyte subsets in the lung [including Mono_CD31_ADJ, Mono_PAI-1_ADJ (but not Mono3)], and all macrophage subsets showed high Jaccard similarity index with HLA DR^hi classical monocytes in the blood (Fig. 7b).

We next interrogated the markers for these two COMBAT cell types (GZB^neg CD8 T cells and HLA DR^hi classical monocytes) (data found in Supplementary Data 3 in COMBAT). We observed that compared to healthy and disease controls, GZB^neg CD8 T cells expressed markers of exhaustion and were KLRG1⁺ compared to other CD8 T cells. HLA DR^hi classical monocytes showed high expression of CLA. Both GZB^neg CD8 T cells and HLA DR^hi classical monocytes were unique amongst CD8 T cell and monocyte subsets in showing lower abundance in COVID-19 patients compared to healthy volunteers[6], raising the possibility that these were the subsets that have trafficked to the lungs. This is not unprecedented given previous findings in lungs which showed sparse antigen-specific T cells in blood of severe influenza patients but 8 times higher in matched blood-lung samples[34].

For neutrophil comparisons between lungs (COSMIC) and blood (COMBAT), we obtained stored whole blood samples and stained these with a 42-marker CYTOF panel (Supplementary Table 5). 8 subclusters of neutrophils were evident from dimensionality reduction (UMAP) and unsupervised clustering, and annotated according to maturity – from pro-neutrophil to mature neutrophils (Fig. 7c, d). Compared to the lung neutrophils, 'immature neutrophil 2' in the blood (which expressed the highest level of CD172a amongst the immature CD10⁻ neutrophil subsets), most closely matched the neutrophil subset in Neut_CD8_ADJ (Fig. 2f). Notably, the abundance of 'immature neutrophil 2' correlated positively with severity of disease (Fig. 7e). These findings showed that the lung CD8 T cell subsets matched most closely to a GZB^neg KLRG1⁺ CD8 T cell subset in the blood, which

also expressed a T cell exhaustion signature. This suggests that this blood CD8 T cell subset is a likely source for the GZB⁺ CD8 T cells found in the Neut_CD8_ADJ cluster; and that within this cluster, CD8 T cells expressed GZB, possibly with exposure to IFN-β[35]. On the other hand, blood CD172a^hi immature neutrophil subset is the likely source for the immature neutrophils in the lungs, including that found in the Neut_CD8_ADJ cluster.

## Discussion

In this paper, we deconvoluted a highly disordered immune and structural landscape to provide accurate annotations and abundance metrics for the cellular landscape and then leveraged mathematical techniques to describe co – location and cell contact-based network construction. Our mathematical tools encompassed a range of spatial statistics and methods from network science; some transposed from ecology[36–38]. The pipeline uncovered a hitherto undescribed physical partnership between immature neutrophils and CD8 T cells in COVID-19 lungs linked to proliferating alveolar epithelium in areas with diffuse alveolar damage. This further connected with classical monocytes and megakaryocyte around endothelial cells, forming a super proinflammatory network across the alveolar bed in DAD. The observations on neutrophils are especially significant since relatively little is understood of the role of neutrophils in the lungs of patients with COVID-19 due to poor detection with transcriptomic methods[17,39].

Our study did not elucidate how neutrophil-CD8 clustering might contribute to disease pathogenesis. However, evidence from other diseases provide some insight. Neutrophils and CD8 T cells aggregation in colorectal cancer and graft vs host disease have been shown to enhance T-cell receptor–triggered activation of CD8⁺ T cells[40] causing neutrophil-mediated tissue damage by the release of reactive oxygen species[41]. Neutrophils can also act as antigen presenting cell which cross present antigen to CD8 T cells, further enhancing activation[42,43]. CD8 T cells with a similar effector memory and GZB⁺ profile as that found in the Neut_CD8_ADJ cluster have also been implicated in immunopathology of COVID-19 in other organs. Imaging mass cytometry studies in COVID-19 brain tissue showed intriguing spatial associations with microglia, which also sustained immune activation and neuroinflammation[44].

The presence of viral antigen could be the trigger for these foci of immature neutrophils and CD8 T cells, possibly initiated by recognition of viral antigen by CD8 T cells. However, we note abundant Neut_CD8_ADJ cluster in the OP state (Fig. 3b) where there were no viral proteins or RNA. One explanation is that these CD8 T cells were self-proliferating, as suggested by Liao's study using single cell RNA sequencing of lung-lavaged cells in COVID-19 patients[45]. Supporting

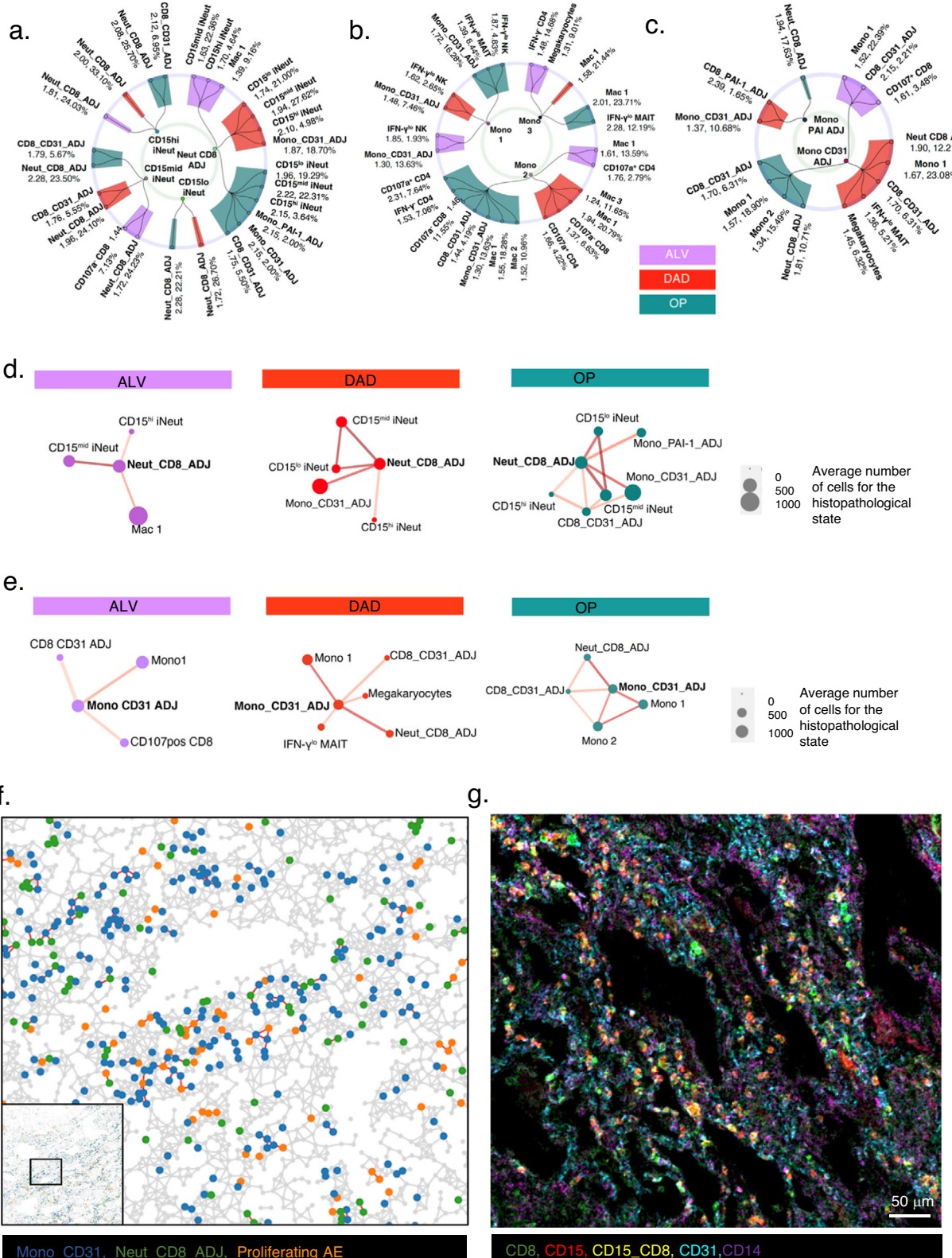

this, Neut_CD8_ADJ cluster showed the highest Ki67 expression (Fig. 2f), with MCD imaging isolating this expression to CD8 T cells (Fig. 2I). Organising pneumonia is not a natural sequela of all viral infection or alveolar inflammation. Indeed, many patients who do well do not progress to consolidation on computed tomographic (CT) scans. Thus, a potential deleterious effect of these foci of inflammation could be the obliteration of regenerative potential in type II alveolar epithelial cells, the purported stem cells for the alveolar unit[46], and development of organising pneumonia (OP).

Another cluster highlighted by our analyses was the Mono_-CD31_ADJ cluster, which was spatially associated with Neut_CD8_ADJ cluster, and with proliferating alveolar epithelial cells. Proliferating alveolar epithelial cells are the nominal stem cells for the alveoli and key to replenishment of type 1 alveolar epithelial cells. Its health, and

**Fig. 6 | Spatial organization amongst immune cells in COVID-19 lungs.** Radial connectivity map depicting all statistically significantly co-located pairs of immune-immature neutrophil subsets (including ADJ subsets) (**a**) immune-monocyte subsets (**b**, **c**, separated for clarity) cells in all histopathology states ($n = 10$ ALV, $n = 8$ DAD and $n = 8$ OP). Anchor cells (immature neutrophil and monocyte subsets) are in smaller, inner circle. Numerical values indicate g(r = 20) for that pair in that state (coloured bar), and % indicates proportion of anchor cells that are co-located with the specified immune cells. These significantly co-located pairs of cells are derived from $n = 479,349$ single cells in all ROIs from $n = 12$ COVID patients' lung sections ($n = 10$ ROIs for ALV; $n = 8$ DAD; $n = 8$ OP); $n = 144,937$ cells in ALV, $n = 146333$ in DAD and $n = 163,506$ in OP (see "Methods" for 3-step mathematical algorithm for determining statistical significance of co-location). Spatial connectivity plots for Neut_CD8_ADJ (**d**) and Mono_CD31_ADJ (**e**), showing immune cells that are statistically significant co-located to proliferating alveolar epithelium (designated 'anchor cell') in the three histopathology states (see "Methods" for 3-step mathematical algorithm for determining statistical significance of co-location). Size of nodes (filled-in circle) represent mean cell counts for the specified cell cluster for all

the ROIs in the histopathology state, and colour of nodes relate to histopathology state. Connecting lines indicate a statistically significant co-location between the two cell types derived from QCM and cross-PCF analyses. Thickness of line relate to value of g(r = 20) relative to each pair in the plot – the thicker the line, the higher the g(r = 20) and strength of co-location between the immune cell type and anchor cell. **f** Adjacency cell network (ACN) map showing contact between the Mono_CD31_ADJ cluster, Neut_CD8_ADJ cluster and proliferating alveolar epithelial. Cell segmentation masks generated by DeepCell were used to produce this spatially-embedded network in which nodes represent centres of cell types (e.g. green – Neut_CD8_ADJ cell cluster). Nodes are connected by a line if the corresponding cells in the segmentation mask share a border. **g** MCD image showing CD8 (green), CD15(red) and CD15 and CD8 co-staining (yellow) (representing Neut_CD8_ADJ cell clusters) amidst endothelial cells (CD31 staining in turquoise) and monocytes (CD14 staining in purple) in a lung section with DAD on histopathology analysis. Exemplar ROI is shown for (**f**) and (**g**), out of 26 ROIs stained, from 12 patients ($n = 10$ ALV ROI, $n = 8$ DAD and n = 8 OP).

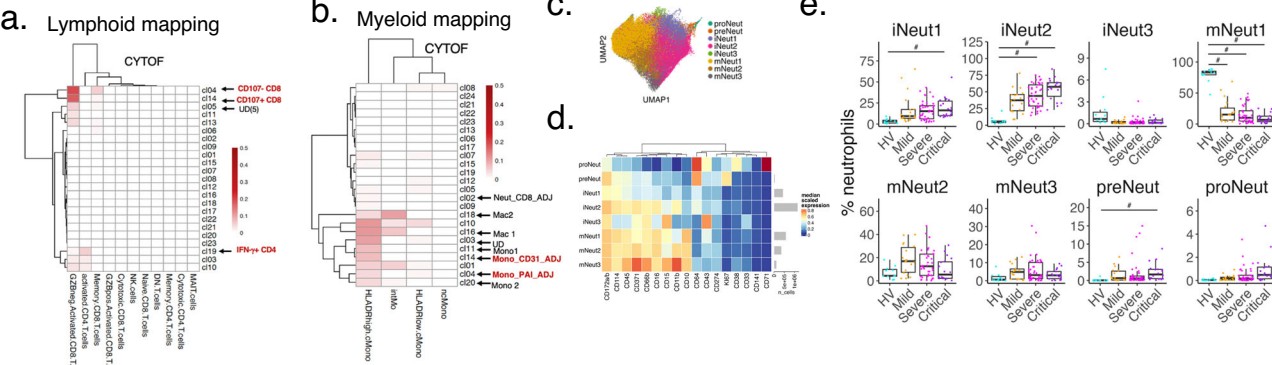

**Fig. 7 | Comparison between blood (COMBAT) and lung (COSMIC) data.**
**a** SCMAP matching heatmaps representing the Jaccard indices of similarity between COMBAT (blood)[6] and COSMIC (lung) lymphocyte clusters. CD107a⁻ CD8 T cell and CD107a⁺ CD8 T cell in COSMIC matched to blood GZB⁻ CD8 T cells in COMBAT. IFN-γ⁺ CD4 T cells matched to COMBAT's 'activated CD4 T cells'. **b** SCMAP matching heatmaps representing the Jaccard indices of similarity between COMBAT (blood) and COSMIC (lung) myeloid clusters. Mono_CD31_ADJ and Mono_PAI-1_ADJ and all macrophage subsets matched with HLA DRʰⁱ classical monocytes in the blood from COMBAT data. **c** UMAP representation of neutrophils from controls and COVID-19 infected patients ($n = 2,776,928$ single cells from $n = 77$ COVID-19 patients and 11 healthy volunteers (HV), down sampled to 100 000 cells per condition) obtained from COMBAT consortium, showing 8 subsets of neutrophils. **d** Heatmap showing median marker expression for genes (selected to match COSMIC's key protein

expression on neutrophils) on the 8 neutrophil subsets, demonstrating high similarity of marker expression in immature neutrophil 2 (iNeut2) in COMBAT (blood) with Neut_CD8_ADJ in COSMIC (lung) (See also Fig. 2f). **e** Abundance of the 8 neutrophil subsets in blood as % of total neutrophils, from healthy volunteers (HV), mild, severe and critical COVID-19 patients from the COMBAT consortium showing a progressive increase in immature 2 neutrophils with increasing COVID-19 disease severity. HV ($n = 11$), mild ($n = 18$), severe ($n = 41$), critical ($n = 18$) patients, $n = 1$ experiment. The boxplot is median, with IQR; whiskers are the range or 1.5*IQR (whichever is smaller). Composition analysis was performed using scCODA with inbuilt adjustment for multiple comparison[70]. Credible compositional changes were identified comparing all groups to HV and FDR < 0.1 is marked with #. Source data are provided in Source Data File.

ability to function optimally, is a key requirement for repair of infected and damaged alveoli. A consequence could be that the production of type I IFN, [and other monocyte-specific cytokines like IL-6 and TNF-α (as reviewed by,[47,48]]), combined to impact on regeneration of alveolar epithelium. It is also possible that type I IFN production from these monocytes causes upregulation of ACE2, thereby sustaining viral entry and alveolar epithelial damage[49]. This agrees with observation from transcriptomic studies of the lungs where type II alveolar epithelium were found in an inflammation-associated intermediate state rather than progressing via normal regeneration to type I alveolar epithelium[13,15,17].

The tight association between a large number of monocytes and endothelial cells in all histopathology states could result in excess inflammation and also predispose to small vessel thrombosis, particularly with further presence of megakaryocytes at the point of maximal injury (DAD) (Fig. 5b). Single cell transcriptomic analyses in COVID lungs have demonstrated upregulation of endothelial-damage markers, including VWF, ICAM1 and VCAM1, and transcriptional programs suggesting altered vessel wall integrity and widespread activation of

coagulation pathway associated genes in endothelial cells[13,16,50]. In addition, autopsy studies have shown high numbers of megakaryocytes and platelet rich thrombi in the lungs with COVID-19 pneumonitis[51].

Beyond these key messages, other findings clarified the importance of immune cell numbers and phenotype in blood of patients with severe COVID-19. There was no significant spatial co-location between activated NK cells and MAIT cells with any structural cells although the numbers for MAIT cells were increased, in keeping with blood levels. With the ability to identify single cells of CD4 and CD8 T cells, and quantify their abundance per mm² of lungs, we also showed definitively that levels of CD4 and CD8 T cells were high in lung samples in contrast to studies which inferred their depletion from gene expression profiles[15]. Immature cycling monocytes, one of the most striking observations in the blood of patients with severe compared to mild COVID-19 disease[6,8], were not found in lung tissue. This suggests that immature monocytes are unlikely to be involved in tissue damage, and unlike immature neutrophils, probably differentiated rapidly to mature monocytes and macrophages.

Our findings refined our earlier work on a smaller subset of COVID-19 lungs ($n = 3$) using targeted transcriptomic analysis (GeoMx) in specified sections in the lungs linked to alveolar damage[52]. In that work, we deconvoluted cells detected by gene expression profile using limited protein markers and showed that CD8 T cells and macrophages with IFN-γ signature correlated with areas of lungs with alveolar damage. Interestingly, areas of severe damage exhibited consistent expression of IFNG-regulated chemokines such as *CXCL9/10/11* that may promote CXCR3-mediated chemotaxis or retention of CD8 T effector lymphocytes. Further to the findings from this paper, we performed additional analyses to determine if we can provide a transcriptomic view of the immature neutrophils and CD8 T cell cluster. This strengthened but did not reveal further findings (described in Supplementary Fig. 11).

Another earlier work in the same lung samples showed significant presence of neutrophil extracellular traps (NETS) in the lung samples which correlated with areas of low CD8 T cell levels. Re-examining the number of NETS per lung section, we observed widespread presence with no significant difference between the three histopathology states (Supplementary Fig. 3C). As NETS production is a feature of mature rather than immature neutrophils[20], one explanation is that there is a CD8-directed immature neutrophil localisation to proliferating alveolar epithelium, which is separate from the relatively less discriminate NETS expression by mature neutrophils.

The key limitation of our study is that it is an observation of association, albeit that there was clear comparison between histopathology characterisations of alveolitis, damage and repair. Thus, it is not possible to elucidate cause or effect. Further functional studies will strengthen the findings. Our cohort was also small though this was counterbalanced by uniquely fresh samples from lungs, with minimal effect of degradation due to the sampling methods at the point of death. Finally, our study was led by specific questions. To that end, the antibody panels, and analyses were targeted to those questions and cellular identities were constrained to that linked to the antibody panel.

We conclude that statistically rigorous analyses of spatial associations of immune and structural cells in lungs of those with fatal COVID-19 identified an inflammatory nidus of immature neutrophils and CD8 T cells with high immune activity and proliferating capabilities that were linked to alveolar progenitor cells in areas with greatest alveolar damage. It establishes the importance of emergency myelopoiesis in lung immune pathology, with potential roles for immature neutrophils and megakaryocytes in alveolar damage, aberrant alveolar regeneration, and excess thrombogenesis. The findings support the evaluation of therapeutics that target monocytes and immature neutrophils, potentially earlier in disease to limit its impact on progression to widespread alveolar damage and organising pneumonia. It also means that drugs that increase the longevity or survival of CD8 T cells require further assessment given the potential contribution of CD8 T cells to lung damage.

## Methods

### Table of antibodies and reagents used in imaging mass cytometry and immunofluorescence
All antibodies, their catalogue numbers, final dilutions, and source are documented in Supplementary Data 1.

### Patients, samples, and ethical approvals
Lung samples were obtained from collaborators from the University of Navarra, Spain and comprised those patients who died in hospital after admission with COVID-19. The only inclusion criteria were (i) hospitalisation, (ii) evidence for COVID-19 pneumonitis, defined as presence of ground glass changes +/− consolidation and peri bronchial shadowing in mid to peripheral distribution on thoracic CT scan begore death, (iii) PCR+ results for nucleocapsid (N) and/or envelope protein I

in lung or liver tissue sample and (iv) negative bacterial culture from blood and lung within 3 days of death. The study was approved by the Ethics Committee of the University of Navarra, Spain (Approval 2020.192). Tissue collections were obtained with consent from a first-degree relative, following a protocol approved by the ethics committee of the University of Navarra (Protocol 2020.192p); and stored under Spain's Human Tissue Authority regulations. Samples were collected during the first wave of pandemic (2020) via an intercostal space incision, using core biopsy methods (BioPince Full Core Biopsy Instrument kit) immediately after death[20,53]. Tissues were immediately fixed in neutral buffered formalin for over 24 hours, and then paraffin-embedded. These samples were also shared with other collaborators and studies carried out independently[20,52].

Healthy lung controls were obtained from the Oxford Centre for Histopathology Research and the Oxford Radcliffe Biobank based at the Oxford University NHS Hospitals Foundation Trust. Ethics approval was received from Oxford A South-Central NHS REC (ref. 19/SC/0173). The inclusion criteria were that lung sections had to be obtained away from localised lung cancer site on lung imaging; they had to have normal lung histopathology as agreed by two independent histopathologists, aged between 50-90 y and had no concomitant lung diseases. Altogether 8 such patients were identified, their lung sections stained with H & E and two representative patients selected to proceed to IMC staining. H & E stained sections are shown in Supplementary Fig. 4.

We have considered sex balance in selection of samples. There are 5 females and 7 males in our cohort. Patients and relatives were not financially compensated.

### RNA extraction and quantitative RT-PCR for viral genes
RNA extraction from biopsies was performed using the QIAamp Viral RNA Mini Kit (Qiagen) and the identification of SARS-CoV-2 transcripts encoding nucleocapsid (N) and an envelope protein I was performed using a commercial kit (SARS-CoV-2 Real Time PCR Kit, Vircell), both according to manufacturer recommendations, at the Microbiology Laboratory of the Clinica Universidad de Navarra (ref). Samples with amplification of both targets with Ct values below 35 were considered positive for SARS-COV-2. Ct threshold was selected based on comparison between Ct values and presence of viral DNA on nasopharyngeal-swab standards.

### SARS-CoV-2 nucleocapsid protein staining
Slides were deparaffinised and heat-induced epitope retrieval were performed on the Leica BOND-RXm using BOND Epitope Retrieval Solution 2 (ER2, pH 9.0) for 30 minutes at 95 °C. Staining was conducted with the Bond Polymer Refine Detection kit, a rabbit anti-SARS-CoV-2 nucleocapsid antibody (Sinobiological; clone: #001; dilution: 1:5000) and counterstained with haematoxylin.

### Region of interest (ROI) selection
H&E stained sections were examined by two senior pathologists independently and a pulmonologist and data compiled with consensus at the third iteration. ROIs were selected based on size (2x 2 mm squares or equivalent surface areas) to represent the dominant histopathology findings for the section. Slides were imaged on AxioScan Z1 slide scanner [Zeiss] and viewed using QuPath[54]

### Imaging mass cytometry (IMC) staining
Sequential 6 μm thick FFPE lung tissue section slides were incubated for 2 hours at 60 °C on a slide warmer, dewaxed twice in Histo-clear II (National Diagnostics) for 10 minutes before rehydration through serial alcohols; 100%, 100%, 95%, 70% ethanol and MilliQ water. Slides were then incubated for 30 minutes at 96 °C in EDTA Target Retrieval Solution, pH 9 (Agilent) and cooled to 70 °C before washing twice in MilliQ water. Slides were blocked in 3% BSA solution in Maxpar PBS

(Standard BioTools; previously Fluidigm) for 45 min. Sections were then stained with metal-conjugated antibodies in Maxpar PBS containing 0.5% BSA overnight. Antibodies conjugated in house were conjugated with MaxPar X8 antibody labelling kits (Standard BioTools) or Lightning-Link kits (Abcam) according to manufacturer's instructions. Slides were washed in 0.2% Triton X-100 then twice in Maxpar PBS. Intercalator-Ir (Standard BioTools) diluted in Maxpar PBS was used to stain DNA (30 min), slides were washed in MilliQ water then air dried.

Ablation of the relevant regions of interest (ROIs) was carried out on Standard BioTools Hyperion Imaging System using CyTOF7 Software v7.0 (Standard BioTools) and visualized using MCD Viewer (Standard BioTools). Images were processed for publication using FIJI[55] to de-speckle and sharpen the images.

### Antibody validation and optimization

Antibody clones were selected which had previously been published and validated in IMC studies as well as antibodies frequently utilized for immunofluorescence or immunohistochemistry studies with FFPE tissues. Staining validation for IMC markers was performed in healthy control lung and tonsil as well as in some COVID-19 infected lung (Supplementary Fig. 5,6 and 12). During optimisation, we checked that (i) mutually exclusive expression pattern were found in key immune and structural lineage markers i.e. CD68, Epcam, CD3 and CD19 (ii) markers showed appropriate subcellular location expression i.e. transcription factors Foxp3 and Ki67 were nuclear, whereas CD68 expression was cytoplasmic and cell membrane. (iii) structural cell identities defined by IMC lineage marker expression are compatible with cell morphology and location in H&E. Adjacent H&E-stained slides and structural markers expression was examined e.g. α-SMA expression around vessels and bronchi, EpCAM expression on bronchial and alveolar epithelial cells. (iv) Non-biological sense expression e.g. CD4 and CD8 co-expression and biologically expected and coherent co-expression patterns eg. cells expressing CD45, CD3, CD8 and CD45RO were examined (v.) Expression for the following key markers was validated by immunofluorescence staining in adjacent slides – CD4, CD8, CD14, CD15, CD31, CD172a, CD206, ProSPC, PAI-1, Epcam and Ki67.

Antibody clones that did not perform well i.e. those with weak signal, high background, or nonspecific staining were discarded. Antibody titration was performed to maximise signal to noise ratio in both lung and tonsil tissues and panels were designed to minimise the already low levels of signal spill over see in IMC [less than 1·5%][56].

### Immunofluorescence

Paraffin-embedded human lung tissue sections were deparaffinized and each section was pre-treated using heat-mediated antigen epitope retrieval with sodium citrate buffer (pH 6) for 20 minutes. Then sections were blocked in 10% normal goat serum (Thermo Fischer Scientific, 50062Z) for 20 minutes and then incubated with CD14 antibody 1:100 dilution (Abcam, AB183322), CD15 antibody 1:200 dilution (Cell signalling Technology, 4744 S), CD31 Antibody 1:100 dilution (LS Bio, LS-B15507-LSP), CD8 Antibody 1:100 dilution(Cell signalling Technology, 90257SF), CD172a, Anti- SIRP-Alpha Antibody 1:100 dilution (Abcam AB19149), Pro-Surfactant Protein C Antibody 1:100 dilution (Abcam AB90716), overnight at 4°C. Each section is washed three times in TBS-T (0.1% Tween) and stained with Alexa Fluor 568 or 647 conjugated Goat anti Rabbit IgG or Alexa Fluor 488 or 568 conjugated goat anti-mouse IgM secondary antibody or Alexa Fluor 488, 568 or 647 conjugated goat anti-mouse IgG1 for 30 minutes and washed three times in TBS-T (0.1% Tween) and mounted with Prolong platinum antifade Mountant with DAPI (Fischer Scientific) and the section slides were imaged using a Nikon Ti2 microscope (Nikon Instruments, Japan) attached to an Andor Dragonfly 200 spinning disk confocal microscope (Oxford Instruments, Belfast).

### Imaging of fluorescent labelled tissue sections

Slides were imaged using a Nikon Ti2-E microscope (Nikon Instruments, Japan) attached to an Andor Dragonfly 200 spinning disk confocal unit (Oxford Instruments, Belfast). Using Andor Fusion software, the microscope was configured for DAPI (Excitation 405 nm: Emission 450/50 nm), GFP (Excitation 488 nm: Emission 525/50 nm), Red (Excitation 561 nm: Emission 600/50 nm) and Far Red (Excitation 647 nm: Emission 700/75 nm). A 10×0.45 NA objective was initially selected to provide an overview of the entire area of the tissue section. Relevant areas (or the whole section) were then selected using the software for higher resolution scanning, utilizing either a Nikon Plan Fluor 40×1.3 NA oil objective with 1 um z-slice sectioning or a Nikon Plan Apo Lambda 100×1.45 NA oil objective with 0.13 um z-slice sectioning, this ensured that the whole thickness of the tissue would be imaged. Images were saved on a computer for further processing using custom Fiji/Image J macros[55].

### Targeted transcriptomic analysis of specific areas of interest with matched IMC staining and analyses

We extracted the RNA sequence data from AOIs ($n = 46$) in three COVID lung sections as described in our previous paper (Cross, A.R. et al. 2022) and organised these into enhanced histopathology classification as described in this paper – ALV, DAD and OP. We then compiled the differential expressed gene list between the three states (using DESeq2) and performed a pathway analysis using Reactome[57] (Supplementary Fig. 11). Here, we found upregulation of genes associated with neutrophil activation when comparing DAD to OP and ALV as observed in Cross A.R. et al. In particular, S100A8 (highly expressed in neutrophils and a feature of degranulation) and CXCL10 (chemokine related to neutrophils trafficking) were highly upregulated, supporting trafficking of neutrophil to the tissue at the DAD phase[58]. High expression of CXCL9 a key chemokine in T cell extravasation into tissue supports finding of T cells (e.g. CD8 T cells) in these AOIs.

### Data analysis

**Software and algorithms.** All software and algorithms used are documented in Table 1.

### The Spatial Omics Oxford (SpOOx) Analysis Pipeline

The SpOOx pipeline is a computational framework that brings together the methods we have used to derive final spatial interpretation for the COVID-19 lung sections. It incorporates a suite of Python and R based command line tools which may be run individually or as a semi-automated pipeline. We have implemented SpOOx using the Ruffus framework[59]. Ruffus allows encapsulation of the workflow and parameters to enable reproducibility, transparency and code reuse. All steps discussed in the Methods are encapsulated in the SpOOx pipeline and example commands to achieve the step are shown below. An overview of the pipeline can be found in Fig. 1a and Supplementary Fig. 1. Full detailed documentation and a tutorial are included on the SpOOx GitHub page (https://github.com/Taylor-CCB-Group/SpOOx). SpOOx produces a series of output directories and files that may be uploaded to the Multi-Dimensional Viewer (MDV) software (see below). MDV has been developed based on the Multi Locus View[60] framework and has been heavily modified and extended to allow visualisation and analysis of large multidimensional data sets, images and the resulting spatial statistics. The code to upload data to MDV is available on GitHub at https://github.com/Taylor-CCB-Group/MDV. Both the SpOOx and MDV are open source under the GPL 3.0 license with these links – SpOOx is available for install at https://github.com/Taylor-CCB-Group/SpOOx and MDV at https://github.com/Taylor-CCB-Group/MDV. The project data analysis is available online within MDV at https://mdv.molbiol.ox.ac.uk/projects/hyperion/6567.

**Table 1 | Software and algorithms used in data analysis**

| Name of software | Source | Identifier |
|---|---|---|
| imctools | https://github.com/BodenmillerGroup/imctools | RRID:SCR_017132 |
| Deepcell | https://vanvalen.github.io/about/ | RRID:SCR_022197 |
| Phenograph | https://github.com/JinmiaoChenLab/Rphenograph | RRID:SCR_016919 |
| Harmony | https://github.com/slowkow/harmonypy | RRID:SCR_022206 |
| Slingshot | https://github.com/kstreet13/slingshot | RRID:SCR_017012 |
| Ruffus | http://www.ruffus.org.uk/ | RRID:SCR_022196 |
| QuPath | https://qupath.github.io/ | https://doi.org/10.1038/s41598-017-17204-5 |
| MCD | https://www.standardbio.com/products-services/software | RRID:SCR_023007 |
| Catalyst R | http://bioconductor.org/packages/CATALYST/ | RRID:SCR_017127 |
| Harmony | https://github.com/immunogenomics/harmony | RRID:SCR_022206 |
| diffcyt R package (version 1.8.8) | https://www.bioconductor.org/packages/release/bioc/html/diffcyt.html | RRID:SCR_023006 |

**Conversion of MCD files to TIFF.** MCD files were checked for problems with ablation or staining using the MCD viewer (provided by Standard BioTools). Once these initial checks were completed, the images were converted to OME-TIFF format for segmentation.

*Commands: python hyperion_pipeline.py make mcd_to_tiff and python hyperion_pipeline.py make tiff_to_histocat*

**Segmentation and cell mask generation**

Cell segmentation was performed with the Mesmer library in DeepCell[61], Nuclear markers (DNA1 and DNA3) and cytoplasmic markers (a-SMA, CCR2, CCR6, CD107a, CD10, CD114, CD115, CD14, CD15, CD16, CD172a, CD31, CD3, CD45, CD45RO, CD4, CD71, CD8a, Collagen1, DAP12, EpCAM, GZB, HLA DR, IFN-β, IFN-γ, PAI1, PanCK, PF4 and RAGE) were extracted to TIFF files and Z projected to single channel nuclear and cytoplasmic single TIFF images (Supplementary Fig. 13). These images were contrast adjusted (--contrast 5) and passed to the Mesmer library (pixel size adjusted to 1 micron) as nuclear and cytoplasm channels. From these, cell segmentation masks were generated for each ROI.

*Command: python hyperion_pipeline.py make deepcell*

**Extraction of signal intensities for each cell**

The intensity of each marker within each labelled cell was extracted from the data using the segmentation masks using the mean arcsinh-transformed (with --cofactor 5) pixel intensity for each. The data were recorded as a table, each row representing a cell with a unique id for the ROI. Shape features such as area, perimeter, eccentricity, and centroid were also extracted from the masks. All cells were then filtered using a cell area greater than 50 μm and less than 300 μm to exclude poorly segmented cells and cell debris. Further QC was performed within MDV by plotting the distribution of marker intensity across each ROI.

*Command: python hyperion_pipeline.py make signal_extraction*

**Dimensionality reduction and cluster analysis**

For all downstream analysis the intensity values were arcsinh transformed with a cofactor 5. Clustering was performed using the Phenograph algorithm[62] through the implementation of the Rphenograph R package (version 0.99.1) with parameter k = 30. Using MDV, the clusters were first visualised using interactive UMAP scatter plots and heatmaps (showing the median marker intensities per cluster) then manually annotated to define the cell phenotypes at the cell level. The clustering was performed at two levels: a sample level (on the trimmed [q = 0.001] and scaled values) and per condition after having integrated the data with Harmony (version 1.0)[63], using the default parameters with the option do_pca = TRUE. The integration of the data was performed per condition to remove variation from different patients and to better define common populations of cells. The annotations

before and after integration were compared to ensure that no biologically meaningful populations were missed when integrating the data. The heatmaps, PCA and UMAP plots were done using the functions from the CATALYST R package (version 1.16.0).

*Command: python hyperion_pipeline.py make phenoharmonycluster*

**Annotation workflow**

Cells were first examined for antibody staining and those cells that did not show any antibody staining were filtered from further analysis. The remaining cells were grouped into three mega-clusters termed Structural, Myeloid or Lymphocyte based on presence and/or absence of CD45, EPCAM, PanCK, CD31, α−SMA, CD56, Vα7.2, CD3, CD14, CD68, PF4 and CD15 expression. The three mega-clusters were then re-clustered using protein markers selected on immunological basis (Supplementary Fig. 7). The resultant final clusters were annotated using an integrated approach. In the first step, we defined clusters using (i) heatmaps showing median marker expression (ii) expression density histograms which allow better delineation of the range of marker expression, specifically differentiating low and negative expression levels and (iii) cluster distribution plots which showed the frequency of each cluster in different samples. Phenotypic similarity of clusters was interrogated via UMAP and cluster dendrograms. To further define cluster identities, the spatial location of clusters was visualised using cell centroid plots and mapped onto an adjacent H&E slide with the same ROI. Based on these analyses, some clusters were excluded under the following criteria: (a) clusters with uniformly low/negative expression of markers, (b) clusters only found in one sample, and (c) Undefined clusters (where the combination of markers did not amount to a subset which could be defined). These clusters were not submitted for spatial analysis. Sub-clusters with very similar expression profiles were merged and those which contained 2 or more clusters were annotated as such. A small number of clusters demonstrated expression of markers normally associated with disparate cell populations (e.g. Neutrophil_CD8 adjacent), which can be attributed to closely apposed cell types. These adjacent cell populations were validated via high resolution immunofluorescence microscopy. To aid final annotation, we also performed Pseudotime inference for selected populations.

Final annotated clusters were then sense-checked against the MCD images by an independent investigator not involved in annotating the clusters, and some key clusters of interest were further examined by immunofluorescence staining with confocal microscopy.

**Pseudotime analysis**

The Pseudotime analysis was performed on the macrophage, monocytes and neutrophils populations (Supplementary Fig. 6). Their arcsinh transformed values were integrated using Harmony with the same parameters as in the main analysis, followed by dimensionality

reduction using UMAP. Then the Pseudotime inference was performed by applying the Slingshot algorithm[64] to the UMAP dimensions using the default parameters and the above annotations as clusterLabels.

This analysis is not part of the SpOOx pipeline but code is available in GitHub.

*Command: R slingshot.R <parameters>*

### Differential cell abundance analysis
Differential abundance analysis between conditions was performed using code from the diffcyt R package (version 1.8.8) with the option testDA_edgeR. To account for the differences in area between the ablated samples, the area was used as a normalising factor. The dispersion was estimated using the option trend.method = "none" and the negative binomial generalized log-linear model was used for the analysis (with the glmFit and glmLRT functions). The BH (Benjamini-Hochberg) method was used to adjust p-values for multiple testing.

### Cell centroid maps
For each ROI, the cell centroids were plotted and coloured according to cell type to produce a cell centroid map which forms the basis of subsequent analyses. These were overlaid with ROI images in MDV so cell types may be located by colour.

### Spatial analyses
A suite of mathematical tools for spatial analyses is incorporated in SpOOx (see below under QCM, cross-PCF and ACN). The following command runs all the spatial analysis methods in SpOOx:

*Command: python hyperion_pipeline.py make spatialstats*

It is also possible to run each spatial function separately and to adjust parameters (see https://github.com/Taylor-CCB-Group/SpOOx/tree/main/src/spatialstats for details). The command line option that can be appended to the basic command above is stated after each method is described.

### Quadrat Correlation Matrix (QCM)
The "Quadrat Correlation Matrix" (QCM) describes correlations between counts of different cell types within square quadrats with edge length 100μm (resulting in between 100 and 400 quadrats per ROI), following an approach used by[38] to identify statistically significant co-occurrences ($p < 0.05$) and applied to multiplex images of cancer by[65].

We construct the QCM by first generating a matrix $\boldsymbol{O}$ whose entries $O_{ij}$ record the number of cells of type $i$ in quadrat $j$, for $1 \leq i \leq n$ and $1 \leq j \leq m$, where $n$ is the number of cell types in the ROI and $m$ is the number of quadrats. We use $\boldsymbol{O}$ to generate 1000 matrices $\boldsymbol{N}^1, \ldots, \boldsymbol{N}^{1000}$ which form a distribution of "observations" in which the number of cells of each type and the number of cells in each quadrat are the same as in $\boldsymbol{O}$, but spatial correlations between cell types are removed by shuffling cell labels. Each matrix $\boldsymbol{N}^k$ is such that, for each $j$:

$$\sum_i N_{ij}^k = \sum_i O_{ij}, \qquad (1)$$

and for each $i$:

$$\sum_j N_{ij}^k = \sum_j O_{ij}, \qquad (2)$$

We construct each matrix $\boldsymbol{N}^k$ as follows. We fix $\boldsymbol{N}^{k,0} = \boldsymbol{O}$, and define rules which permute the entries of $\boldsymbol{N}^{k,s}$ to obtain a new matrix $\boldsymbol{N}^{k,s+1}$. This is accomplished by selecting two rows *(a,b)* and two columns *(c,d)* of $\boldsymbol{N}^{k,s}$ at random. For some integer $p$ sampled uniformly at

random from the interval $[0, \min(N_{bc}^{k,s}, N_{ad}^{k,s})]$, we then fix:

$$N_{ac}^{k,s+1} = N_{ac}^{k,s} + p, \qquad (3)$$

$$N_{bc}^{k,s+1} = N_{bc}^{k,s} - p \qquad (4)$$

$$N_{bd}^{k,s+1} = N_{bd}^{k,s} + p \qquad (5)$$

and

$$N_{ad}^{k,s+1} = N_{ad}^{k,s} - p. \qquad (6)$$

This process is repeated for $s = 0, 1, \ldots 10,000$ to ensure that the final matrix $\boldsymbol{N}^k = \boldsymbol{N}^{k,10000}$ is well shuffled.

Partial correlation matrices $\boldsymbol{C_O}$ and $\boldsymbol{C}_{N^1}, \boldsymbol{C}_{N^{1000}}$ are then calculated for $\boldsymbol{O}$ and $\boldsymbol{N}^1, \ldots, \boldsymbol{N}^{1000}$ respectively. Standard effect sizes (SES) are determined by rescaling the partial correlations in $\boldsymbol{C_O}$ by the element-wise mean $\mu$ and standard deviation $\sigma$ of the $\boldsymbol{C}_{N^k}$, such that

$$SES_{ij} = (\boldsymbol{C}_{O_{ij}} - \mu[\boldsymbol{C}_{N^k}]_{ij})/(\sigma[\boldsymbol{C}_{N^k}]_{ij}) \qquad (7)$$

Non-significant associations are identified by calculating a 2-tailed *p*-value for each pair of cell types and applying a Benjamini-Hochberg correction, with false discovery rate FDR = 0.05. Non-significant entries of SES are set to 0 in order to generate the QCM, a cell association matrix whose non-zero entries identify standardised effect sizes of pairs of cell types that are statistically significantly correlated within the ROI.

The average QCM across $Q$ ROIs is obtained by concatenating the relevant observation matrices. Denoting by $\boldsymbol{O}_q$ the observation matrix from *ROI q*, we concatenate $\boldsymbol{O}_1, \ldots, \boldsymbol{O}_Q$ to form a combined observation matrix $\boldsymbol{O} = (\boldsymbol{O}_1 \boldsymbol{O}_2 \boldsymbol{O}_Q)$, an $(n \times (m_1 + m_2 + \ldots + m_Q))$ matrix, where $m_q$ denotes the number of quadrats in *ROI q*. Similarly, we concatenate $\boldsymbol{N}_1^k, \ldots, \boldsymbol{N}_Q^k$ to form $\boldsymbol{N}^k = (\boldsymbol{N}_1^k \boldsymbol{N}_2^k \boldsymbol{N}_Q^k)$. Standard partial correlation matrices are then calculated and then the process described above for a single ROI is used to compute the average QCM for multiple ROIs.

*Command option: --function morueta-holme*

### Cross pair correlation functions (cross-PCF)
Significant correlations identified at length scales in the range 0-100μm via the QCM are further assessed by using cross pair correlation functions (cross-PCFs – see, e.g., Bull 2020). Cross-PCFs quantify clustering and dispersal of pairs of cell populations across a range of length scales (here 0-300μm). The cross-PCF considers pairs of cells which are separated by distances $r \in [r_k, r_{k+1})$, where $r_0 = 0$ and $r_k = r_{k-1} + 10$ for $k = 1, \ldots, 30$.

For cell populations A and B, the cross-PCF, $g(r_k)$, is defined as follows:

$$g(r_k) = \frac{1}{N_A} \sum_{a=1}^{N_A} \sum_{b=1}^{N_B} \frac{I_{[r_k, r_{k+1})}(|\boldsymbol{x}_a - \boldsymbol{x}_b|)}{\rho_B A_{r_k}(\boldsymbol{x}_a)}, \qquad (8)$$

where $N_A$ and $N_B$ are the numbers of cells of types A and B, $A_{r_k}(\boldsymbol{x})$ is the area of that portion of an annulus centred at $\boldsymbol{x} = (x,y)$ with inner radius $r_k$ and outer radius $r_{k+1}$ which falls within the ROI, $\boldsymbol{x}_a$ and $\boldsymbol{x}_b$ are the spatial coordinates of cells $a$ and $b$ (of types A and B respectively), $I_{[r_k, r_{k+1})}(r)$ is an indicator function ($I_{[r_k, r_{k+1})}(r) = 1$ if $r \in [r_k, r_{k+1})$ and $I_{[r_k, r_{k+1})}(r) = 0$ otherwise), and $\rho_B$ is the density of cells of type B in the ROI.

A cross-PCF with $g(r) > 1$ means that cells of type A are observed more frequently at distance r from cells of type B than would be expected under complete spatial randomness (CSR), and is indicative of clustering at distance $r$. Conversely, a cross-PCF with $g(r) < 1$ means that cells of type A are observed less frequently at distance r from cells

of type B than would be expected under CSR, and is indicative of exclusion.

For individual ROIs, 95% confidence intervals are obtained via bootstrapping. The spatial dependence of resampled points is accounted for by resampling grid sites within a 20μm square lattice, following[66].

To aid comparison between the clustering and dispersal of different pairs of cell populations, we frequently report cross-PCF values at $r_k = 20$, corresponding to length scales in the range $r \in [20,30)$ μm. We focus on $r_k = 20$ since it approximates the distance between the centroids of cells which are in physical contact. For notational simplicity, we denote this value as $g(r=20)$.

*Command option: --function paircorrelationfunction*

## Topographical correlation map

The cross-PCF quantifies clustering and dispersal of pairs of cell populations at different length scales within an ROI. We also introduce the Topographical Correlation Map (TCM), to visualise how the spatial correlation between cells of types A and B, say, changes across an ROI.

In order to define $\Gamma_{ab}$, the TCM for cells of types A and B, we first associate a mark $m_{ab}$ with each cell $a$ of type A. The mark $m_{ab}$ is defined to be the ratio of $b$, the number of cells of type B within 100μm of cell $a$, to the expected number of cells of type B if they were distributed according to CSR:

$$m_{ab} = \sum_{j=1}^{N_B} \frac{I_{[0,100)}\left(\left|\boldsymbol{x}_a - \boldsymbol{x}_j\right|\right)}{\rho_B A_{100}(\boldsymbol{x}_a)}, \tag{9}$$

where $\rho_B$ is the density of cells of type B in the ROI, $A_{100}(\boldsymbol{x}_a)$ is the area of that portion of a circle with radius 100μm centred at $\boldsymbol{x}_a = (x_a, y_a)$ which falls within the ROI, $I_{[0,100)}(r)$ is an indicator function ($I_{[0,100)}(r) = 1$ when $0 \le r < 100$ and $I_{[0,100)}(r) = 0$ otherwise), and $N_B$ is the total number of cells of type B within the ROI. We interpret values of $m_{ab}$ in a manner similar to that used for cross-PCFs: $m_{ab} < 1$ indicates anti-correlation between cells of types A and B within a distance of 100μm, and $m_{ab} > 1$ indicates correlation.

To facilitate visualization and interpretation, we normalize the mark $m_{ab}$ by introducing the transformed mark, $M_{ab}$, where:

$$M_{ab}(m_{ab}) = 1 \text{ if } m_{ab} \ge \alpha, \tag{10}$$

$$M_{ab}(m_{ab}) = \frac{m_{ab} - 1}{\alpha - 1} \text{ if } 1 < m_{ab} \le \alpha, \tag{11}$$

$$M_{ab}(m_{ab}) = \frac{1 - \frac{1}{m_{ab}}}{\alpha - 1} \text{ if } \frac{1}{\alpha} < m_{ab} < 1, \tag{12}$$

$$M_{ab}(m_{ab}) = -1 \text{ if } m_{ab} \le 1/\alpha. \tag{13}$$

The constant $\alpha$ defines a threshold for extreme clustering. If $m_{ab} > \alpha$ then we have strong clustering and we fix $M_{ab} = 1$; if $m_{ab} \le 1/\alpha$ then we have strong exclusion and we fix $M_{ab} = -1$.

A sketch of $M_{ab}$ is presented in Supplementary Fig. 14.

We note the following properties of the transformed mark, $M_{ab}$. First, $M_{ab}(m_{ab}) = -M_{ab}(\frac{1}{m_{ab}})$, so that dispersal and clustering are measured on the same scales. For example, $m_{ab} = 2$ indicates the presence of twice as many cells of type B as expected under CSR, while $m_{ab} = 1/2$ indicates the presence of half as many cells of type B as expected under CSR. Secondly, the magnitude of $M_{ab}$ describes the strength of the spatial interaction. Finally, the sign of $M_{ab}$ identifies whether there is clustering ($M_{ab} > 0$) or exclusion ($M_{ab} < 0$) between cell $a$ (of type A) and cells of type B.

The parameter $\alpha$ characterises the most extreme clustering or exclusion which can be resolved in each kernel, with extremal values being mapped to 1 and -1 respectively. We use $\alpha = 5$, so clustering or exclusion stronger than 5x is interpreted as the strongest clustering/exclusion that we can distinguish.

After calculating $M_{ab}$ for each cell of type $a$ across the ROI, we centre a Gaussian kernel, with standard deviation $\sigma = 50\mu m$, and maximum height $M_{ab}$, at $\boldsymbol{x}_a$. We sum the kernels associated with all cells of type A to generate the TCM, $\Gamma_{ab}(\boldsymbol{x})$:

$$\Gamma_{ab}(\boldsymbol{x}) = \sum_{a=1}^{N_A} \frac{M_{ab}}{\sigma\sqrt{2\pi}} e^{-\frac{1}{2\sigma^2}|\boldsymbol{x}-\boldsymbol{x}_a|^2} \tag{14}$$

The TCM permits identification of spatial locations in which cells of type A are positively ($\Gamma_{ab} \gg 0$) or negatively ($\Gamma_{ab} \ll 0$) associated with cells of type B. For computational efficiency, when calculating $\Gamma_{ab}$, we assume that each kernel has compact support, being centred in a square region of edge length 300μm.

Finally, we note that $\Gamma_{ab} \ne \Gamma_{ba}$, since the kernels used to construct $\Gamma_{ab}$ are centred on cells of type A (and vice versa). While areas in which cells of type A and type B are co-located should be identified by both $\Gamma_{ab}$ and $\Gamma_{ba}$, their values will differ in regions rich in one cell type and poor in another. We therefore stress that $\Gamma_{ab}$ describes locations in which cells of type A are correlated or anti-correlated with cells of type B, and that the presence or absence of cells of type B cannot be inferred from regions in which $\Gamma_{ab}$ is close to 0.

*Command option: --function localclusteringheatmaps*

## Adjacency cell networks

We use the cell segmentation masks generated by DeepCell to produce a spatially-embedded adjacency cell network (ACN), whose nodes represent cell centres and are labelled according to their cell type. Nodes are connected by an edge if the corresponding cells in the segmentation mask share a border. To ensure that small perturbations in cell boundaries do not lead to errors in cell connections, we expand the border of each segmented cell by 5 pixels before generating the network.

We use the ACN to define two statistics for each pairwise combination of cell types A and B. First, we compute $\phi_{AB}$, the proportion of cells of type A which are in contact with at least one cell of type B:

$$\phi_{AB} = \frac{1}{N_A} \sum_{a=1}^{N_A} I_B(a), \tag{15}$$

where $N_A$ is the number of cells of type A and $I_B(a)$ is an indicator function ($I_B(a) = 1$ if cell $a$ is connected with a cell of type B and $I_B(a) = 0$ otherwise. Secondly, we calculate $\Phi_{AB}$, the average number of cells of type B that are in contact with a cell of type A:

$$\Phi_{AB} = \frac{1}{N_A} \sum_{a=1}^{N_A} \eta_B(a), \tag{16}$$

where $\eta_B(a)$ is the number of cells of type B in contact with cell $a$.

In this paper, we used the ACN to calculate $\phi_{AB}$, the proportion of cells of type A that have at least one cell of type B in contact with them, and $\Phi_{AB}$, the average number of cells of type B that are in contact with a cell of type A in the ROI.

*Command option: --function networkstatistics*

## Multi-dimensional viewer (MDV)

MDV is a comprehensive spatial analytics platform that facilitates the interrogation of large complex data sets and includes various interactive dashboards to facilitate quality control, interactive clustering, phenotyping and spatial analysis. It is an open source web application which can be downloaded and installed locally or used on the publicly

available web site http://mdv.molbiol.ox.ac.uk. Users register to use the site and projects can private, shared with other users or made public. Full documentation and tutorial videos are provided on the MDV website but we provide an overview here.

MDV allows output generated by the SpOOx pipeline to be loaded at different states. Data locations are specified in a yaml format file which can be edited by the user (command: python mdvupload.py myconfig.yaml). Examples of data tables that may be uploaded are:

- Image data (PNGs/OME-TIFF stacks): ROI image stacks, H and E images binary cell masks.
- Cell data (tab separated file): one cell per row, including size, size, shape, phenograph clusters identification, UMAP coordinates, marker signal intensities.
- Spatial Statistics data (tab separated file): one row containing cell to cell interaction data and associated statistics.
- Data related to the disease states (JSON file): allowing grouping of samples for high level analysis.

Once uploaded the data are presented in MDV as a series of views that contain multiple interactive charts corresponding to different analytical methods from clustering, annotation, cell centroid visualisation and spatial analytical methods. Each view focuses on a particular aspect of the pipeline. View contents can be adjusted and added to by adding other chart types and saved as a new view. Chart types can be D3 components (https://d3js.org/) but we have also written custom chart types for performance reasons. For example, MDV scatterplot chart can visualise and interrogate at least 10 million data points. We also integrate Viv viewer[67] to visualise composite image stacks.

The complete analysis and data set were published by publicly sharing the data at https://mdv.molbiol.ox.ac.uk/projects/hyperion/6356.

### COMBAT data mapping

**COMBAT CyTOF data generation and processing.** Cell suspension mass cytometry (CyTOF) data were generated by the COMBAT consortium as previously described[6]. In brief, whole blood from COVID-19 patients was stabilised using a Cytodelics fixative solution, red blood cells were lysed, cellular material was fixed, and samples were run in a Helios CyTOF machine. Importantly, samples were enriched for mononuclear cells before profiling by performing magnetic depletion of CD66+ granulocytes.

After acquisition, data were formatted into a single-cell protein abundance table and annotated into cell types based on marker expression[6]. For the analyses in the present study, this expression matrix was split into two subsets: one containing T and NK cell types, and a second one containing myeloid cell types (i.e. monocyte subsets).

**Mapping cells from lung tissue to blood cells from the COMBAT study[6].** Cells in the lung dataset were matched to the most closely related cell types in blood using scmap, a method which enables label projection by calculating the similarity between cells profiled by two separate studies[33]. In brief, CyTOF and CITE-seq expression matrices from the COMBAT study were used to build index references for label projection. First, proteins which were detected in both studies were identified. This resulted in a panel of 13 and 18 proteins shared between our study and the COMBAT CyTOF and CITE-seq panels, respectively. Next, these proteins were used as a basis for cell type classification with the scmapCluster function. Classification accuracy was tested by splitting the COMBAT data into training and test sets containing 80% and 20% of cells, respectively. The training set was used to generate the scmap reference index, while the test set was used to assess cell type prediction accuracy[33]. Given the reduced set of markers shared between studies, not all COMBAT cell populations could be accurately predicted. Thus, in order to maximise predictive accuracy similar

subpopulations of the same cell type were merged into a single group and any cell types known to be absent from our lung data, such as B cells and plasmablasts, were removed. This approach achieved over 70% accuracy for CyTOF data (71% and 78% predictive accuracy for myeloid and lymphoid cell types, respectively) and 85% accuracy for CITE-seq data and components of final merged clusters are shown in Supplementary Fig. 16.

Indexed references were next used to match cells in the lung to the most similar clusters in blood using the scmapCluster() function. To do so, cell type labels were predicted for each cell in the lung based on the CyTOF and CITE-seq reference sets. Any unassigned cells were discarded. Cluster overlap between studies was visualised using Sankey diagrams[68] and quantified using Jaccard indexes[69].

**Neutrophil subset analysis from stored COMBAT samples.** Whole blood samples frozen in whole blood cell stabilizer (Cytodelics) were obtained from COMBAT consortium storage for healthy volunteers ($n = 11$), health care workers ($n = 12$), COVID-19 ($n = 93$) and Sepsis ($n = 48$). Pre-processed CD45+ gated FCS files of granulocyte containing whole blood samples were analysed with R (v4.0.0). 50000 cells per sample were integrated using Harmony (v1.0)[2] and the CATALYST package (v1.14.0)[3] was used for downstream analysis. CD45+ cells were clustered based on the FlowSOM and ConsensusClusterPLus algorithms using the cluster () function. 50 metaclusters (xdim=100, ydim=100, k = 50) were then assigned to major cell types (T cells, B cells, plasmablasts, mononuclear phagocytes and neutrophils). Neutrophils were selected and reclustered based on CD45, CD15, CD38, CD64, CD16, CD43, CD66b CD10, CD33, KI67, CD172a/b, CD141, CD71, CD114, CD371 and CD274 expression. 30 neutrophil metaclusters (xdim=100, ydim=100, k = 30) were manually merged to 8 neutrophil clusters (proNeut, preNeut, iNeut1-3, mNeut1-3) based on median marker expression.

### Reporting summary
Further information on research design is available in the Nature Portfolio Reporting Summary linked to this article.

## Data availability
The spatial mass cytometry dataset (MCD) files and results of analysis by the Spatial Omics Oxford pipeline are available at https://doi.org/10.5281/zenodo.6513508. The analysis results are also presented as a dynamic online resource in Multi-Dimensional Viewer (MDV) (https://mdv.molbiol.ox.ac.uk/projects/hyperion/6567). All source data are found in https://doi.org/10.5281/zenodo.6513508; and also within the hyperion 6567 project in the MDV link. Specific source data for graphs are also provided in Source Data File. Source data are provided with this paper.

## Code availability
The complete code for the Spatial Omics Oxford pipeline is available as a GitHub repository under the GPL license: https://github.com/Taylor-CCB-Group/SpOOx. In addition, the SpOOx pipeline has been deposited at Zenodo (https://zenodo.org/record/8320986). The Multi-Dimensional Viewer code is available under the GPL license: https://github.com/Taylor-CCB-Group/MDV. This package has also been deposited at Zenodo (https://zenodo.org/record/8324918).

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

## Acknowledgements

We thank patients' relatives for donation of the patients' lung tissue. This work was funded by the Oxford University Medical Sciences Division COVID Funds, NIHR Oxford Biomedical Research Centre and the Chinese Academy of Medical Sciences (CAMS) Innovation Fund for Medical Science (CIFMS), China (grant number: 2018-I2M-2-002). LD and CV are supported by the NIHR Oxford Biomedical Research Centre. LPH is supported by MRC Human Immunology Unit grant (MC_UU_00008/1) and the NIHR Oxford Biomedical Research Centre (NIHR203311).

## Author contributions

P.W. analysed the data, contributed to development of MDV, spatial analyses, and interpretation of mathematical output and all spatial data, and writing of the paper. L.D. contributed to analysis and interpretation of data, optimised and performed the staining of the lung sections in conjunction with RE, analysed MCD images and contributed to the writing of the paper. J.B. performed all the mathematical development and analysis in conjunction with HB and contributed to interpretation of data and writing of the paper. E.R. performed all bioinformatic analysis and contributed to interpretation of data and writing of the paper. C.V. performed all the immunofluorescence and imaging of sections. G.D.H.T. and C.C. performed all the histopathology analyses in conjunction with L.P.H.. A.C. performed the protein immunostaining of the sections and contributed to interpretation of data. C.E.D.A. and I.M.B. organised acquisition of patient samples, clinical data and ethical permissions. Y.X.Z. and D.A. optimised, performed CYTOF experiments on neutrophils and analysed data. E.C.G. and J.W. performed all COSMIC v COMBAT data analysis in conjunction with P.W., L.P.H and J.K.. D.R. and P.K. interpreted data, acquired some of the funding for the study and contributed to writing of the paper. T.D., I.A.U., G.O., C.M., J.K., F.I. interpreted data, discussed annotations and immunological analysis, and contributed to writing of the paper. D.S. and S.M.G. performed early code testing for the pipeline. S.T. oversaw all computational work and code writing for the study, contributed to analysis of data and writing of paper. M.S. performed all the dataset organisation and spatial analysis set up in M.D.V., wrote the codes and organised MDV in conjunction with P.W., S.T. and L.P.H.. J.M.W., A.A., P.W. and L.P.H. performed and analysed all the transcriptomic data from previous studies and publicly deposited data, and revisions of paper. H.B. oversaw all mathematical development and contributed to writing of the paper. L.P.H. conceptualized, led the study, acquired funding, analysed the data and wrote the manuscript. All authors read and approved the final manuscript. Our authorship is diverse, inclusive and equal. There is representation from both sexes, high and low income countries of origin, age, and minorities.

## Competing interests

P.K. has acted as a consultant for Biomunex, Infinitopes, Astra Zeneca and UCB. The remaining authors declare no competing interests.

## Additional information

[1]MRC Translational Immunology Discovery Unit, MRC Weatherall Institute of Molecular Medicine, University of Oxford, Oxford, UK. [2]Wolfson Centre for Mathematical Biology, Mathematical Institute, University of Oxford, Oxford, UK. [3]MRC WIMM Computational Biology Unit, MRC Weatherall Institute of Molecular Medicine, University of Oxford, Oxford, UK. [4]Department of Cellular Pathology and Radcliffe Department of Medicine, Oxford University Hospitals NHS Foundation Trust, Oxford, UK. [5]Anatomic Pathology, Weill Cornell Medical College, Doha, Qatar. [6]Nuffield Department of Surgical Sciences, University of Oxford, Oxford, UK. [7]Navarra Institute for Health Research, Pamplona, Spain. [8]Kennedy Institute for Rheumatology, University of Oxford, Oxford, UK. [9]Wellcome Centre for Human Genetics, Nuffield Department of Medicine, University of Oxford, Oxford, UK. [10]Nuffield Department of Orthopedics, Rheumatology and Musculoskeletal Diseases, University of Oxford, Oxford, UK. [11]Nuffield Department of Medicine, University of Oxford, Oxford, UK. [12]Chinese Academy of Medical Science (CAMS) Oxford Institute (COI), University of Oxford, Oxford, UK. [13]Ludwig Institute for Cancer Research, University of Oxford, Oxford, UK. [14]Respiratory Medicine Unit, Nuffield Department of Medicine, University of Oxford, Oxford, UK. [15]These authors contributed equally: Praveen Weeratunga, Laura Denney, Joshua A. Bull, Emmanouela Repapi. ✉e-mail: Stephen.Taylor@well.ox.ac.uk; Ling-pei.ho@imm.ox.ac.uk

