## [Peer Review File · Nature Communications]

Single cell spatial analysis reveals inflammatory foci of immature neutrophil and CD8 T cells in COVID-19 lungsEditorial Note: Parts of this Peer Review File have been redacted as indicated to maintain the confidentiality of unpublished data.

REVIEWER COMMENTS

Reviewer #1 (Remarks to the Author):

I have reviewed the paper on Unbiased single cell spatial analysis localises active immature neutrophil-CD8 T cell clusters to alveolar progenitor cells in fatal COVID-19 lungs. The paper is a observational spatial biology study on a unique sample cohort and takes advantage of modern multiplexed immunohistochemistry Cytof approach combine with other fluorescent approaches to identify various cell types in post-mortemlung samples from Covid-19 patients. The study is analytically well conducted and offers a unique dataset that by itself merits publication to provide access to the broader research community. While the workflow and associated software is available, I could not find if the entire dataset would be made available with its own digital object identifier. For me this would be essential. While the authors have done an outstanding job in exploring the spatial correlation landscape, the actual molecular signal causality of the observation and characterisations of alveolitis, damage and repair is not yet elucidated in this work. Making the data available would potentially assist other researchers to pursue cause and effect.

One of the main achievements described in the work is the development and evaluation of the spatial biological single cell analysis pipeline based on a newly developed 37-plex antibody panel that resulted in this unique data. While single cell phenotyping and correlation analysis by itself is not necessarily very innovative, the combination of application, data, pipeline and distance correlation maps is definitely novel. Moreover, the authors have quantified spatial correlations based on cellular proximity and thus provide new insights on spatial cellular stressors during in covid affected tissue. It also deomstrates the importance of spatial analyses as diluted circulation samples or tissue homogenates would never have provided this insight.

The reported graphical networks are very insightful, the difference with the healthy controls is striking for the usual marker suspects during inflammation, but as there are only 2 healthy controls employed, I wonder how they were selected. What did theses individuals die from,

whas their age comparable with the age of the Covid victims. In other words, how comparable are these lungs to the Covid lungs and can the authors excluded that the control was itself on an accidental n=2 observation? It would be good to describe and include the tissue selection criteria, which would improve the impact of the paper. Ideally a similarly large control group would be analyse to make the statistics comparable.

Reviewer #2 (Remarks to the Author):

To understand better the immune cell distribution in the COVID-19 lungs, this study the authors developed some mathematical tools (radial connectivity map, topological correlation map) to search for statistically significant co-locations amongst immune and structural cells identified using 37-plex imaging mass cytometry. They have found a highly active cluster of immature neutrophils and cytotoxic CD8 T cells that was spatially linked with alveolar progenitor cells. They have also found that CD8 T cells, monocytes and immature neutrophils are linked to disease severity.

The authors have provided their pipeline and the visual-analytical tool software.

Overall it is an interesting, well-written manuscript that provides new knowledge about the cellular immune landscape in the COVID-19 lung: cell numbers, phenotypes, spatial associations between different immune cell types. Therefore, the manuscript deserves publication.

Reviewer #3 (Remarks to the Author):

The manuscript entitled “Unbiased single cell spatial analysis localises active immature neutrophil-CD8 T cell clusters to alveolar progenitor cells in fatal COVID-19 lungs” by Weeratunga et al, performed a single cell spatial analysis of the immune-structural cell interactions in COVID-19 lungs. The investigators reveal a detailed cellular map, highlight newly identified clusters and offer a pipeline for future spatial analyses.

Recommendation: Redirect to a methods or resources journal eg Nature Methods

Major/General comments:

1. The strength of this work clearly lays on the quality and type of samples utilised, as well as the pipeline developed and made available. As such, this reviewer believes that the paper would suit another type of journal, with a stronger focus on methodology and technical/scientific resources.
2. Indeed, the paper is purely descriptive and the interpretation of any interesting findings such as the presence of “neutrophil-CD8” and “monocyte-endothelial” clusters is at least speculative.
3. Additional spatial single cell omics data from the same specimens could confirm specific interesting findings of this work. For example, the discrepancy between transcriptomic and celomic approaches in regards to conclusions on the role of neutrophils in disease due to their low RNA content.

REVIEWER COMMENTS (in bold) and point-point response

Reviewer #1 (Remarks to the Author):

I have reviewed the paper on Unbiased single cell spatial analysis localises active immature neutrophil-CD8 T cell clusters to alveolar progenitor cells in fatal COVID-19 lungs.

Thank you for your time in reviewing our M/S. Please find a point-by-point response below in bold. Figures are embedded in the appropriate points in this document but provided in higher resolution in the Appendix.

The paper is a observational spatial biology study on a unique sample cohort and takes advantage of modern multiplexed immunohistochemistry Cytof approach combine with other fluorescent approaches to identify various cell types in post-mortem lung samples from Covid-19 patients. The study is analytically well conducted and offers a unique dataset that by itself merits publication to provide access to the broader research community.

Thank you for this positive comment.

While the workflow and associated software is available, I could not find if the entire dataset would be made available with its own digital object identifier. For me this would be essential. While the authors have done an outstanding job in exploring the spatial correlation landscape, the actual molecular signal causality of the observation and characterisations of alveolitis, damage and repair is not yet elucidated in this work. Making the data available would potentially assist other researchers to pursue cause and effect.

Of course. The DOI was found under 'Datasets' just before 'Acknowledgements' (doi:10.5281/zenodo.6513508). This contains the entire dataset which we used for the analysis. The link is now updated and I have pasted it here for your convenience - (<https://zenodo.org/record/6513508/>)

One of the main achievements described in the work is the development and evaluation of the spatial biological single cell analysis pipeline based on a newly developed 37-plex antibody panel that resulted in this unique data. While single cell phenotyping and correlation analysis by itself is not necessarily very innovative, the combination of application, data, pipeline and distance correlation maps is definitely novel. Moreover, the authors have quantified spatial correlations based on cellular proximity and thus provide new insights on spatial cellular stressors during in covid affected tissue. It also deomstrates the importance of spatial analyses as diluted circulation samples or tissue homogenates would never have provided this insight.

Thank you very much for these positive comments.

The reported graphical networks are very insightful, the difference with the healthy controls is striking for the usual marker suspects during inflammation, but as there are only 2 healthy controls employed, I wonder how they were selected. What did theses individuals die from, whas their age comparable with the age of the Covid victims.

Apologies for this oversight. We have now added the clinical information for the healthy control in the text. The two samples were from two men, a 70y and 64y old who had localised lung cancers (age comparable to COVID patients which had a mean (S.D.) age of 72(16)y. Sections were obtained far

from the lobectomy and analysed by histopathologists to show normal lung. We have added a table for this (Table 1 in attached Figures and Tables Appendix).

In other words, how comparable are these lungs to the Covid lungs and can the authors excluded that the control was itself on an accidental n=2 observation? It would be good to describe and include the tissue selection criteria, which would improve the impact of the paper. Ideally a similarly large control group would be analyse to make the statistics comparable.

In terms of describing the tissue selection, criteria – thank you for this request– we have added this to the paper. In essence, the samples from COVID patients were obtained in consecutive patients who died, and whose relatives provided assent, during the first wave of the pandemic. The healthy control lung sections were selected as described above – they have to be normal lung sections (examined by two lung histopathologists independently to each other). For COVID, regions of interest (ROIs) were selected to be representative of the lung section’s histopathology. We have made this clearer in the methods.

In terms of comparability, the sections from our two healthy controls (HC) are highly representative of normal sections of lungs which tend to be uniformly covered with alveolar epithelial lining with occasional blood vessel and respiratory bronchiole. Figure 1 shows a typical healthy lung section on left compared to a typical ROI from COVID pneumonitis in our study. To provide evidence for this assertion, we have acquired healthy control lung sections from a further 6 individuals (demographics in Table 1) to demonstrate this point – Figure 2 shows that the ROIs are near identical to each other from a histopathology perspective and to the two donor’s lungs we used (in blue box).

Figure 1 A. Normal lung H&E section showing thin unicellular alveolar lining (AL) and alveolar air space and scarce immune cells, bronchial blood vessels (BV) with red blood cells (RBC). B. A typical COVID sample for comparison, showing massive infiltration of immune cells, thickened alveolar lining with evidence of alveolar damage and type II alveolar metaplasia, expanded interstitium, fibroblast proliferation and obliteration of airspace (this was a lung section with organizing pneumonia histopathology)

A.

B.

Figure 2. Figure shows the ROIs for HC1 and HC2 which were submitted for spatial analyses in the manuscript (M/S) (in blue box), and ROIs from additional n=6 HC lungs for comparison. All samples show typical thin unicellular alveolar lining (AL) with minimal number of immune cells. In some ROIs, normal broncho-vascular bundle (Br= bronchiole, BV = blood vessel) is evident. Varying sizes of normal blood vessels are seen (smaller – usually smaller pulmonary veins, larger with smooth muscle sheath are pulmonary veins or bronchial arteries). B. Representative histology for lung sections of COVID samples submitted for spatial analyses (original Figure S2 in M/S). Full histopathology description was provided in M/S (Suppl Methods).

In terms of a similarly large group of controls to make this statistically comparable, we would be grateful if the following could be considered:

A. Structural cells

B. Myeloid cells

C. Lymphoid cells

Figure 3 Cell numbers from lung lung sections (HC) from original Figure 2 in original manuscript, showing consistently low number of immune cells (near zero for most).

A striking and common feature of the control samples is the extremely low number of immune cells as is evident in Figure 3 (cut and pasted here from Figure 2 in original M/S for convenience). This is expected in healthy lungs, and as shown in Figure 2A in this document, is consistently and uniformly low across the lung sections from n=8 patients.

In the mathematical pipeline, each ROI is first divided into 100x100um quadrats and correlation in numbers between all pairs of annotated cells in each quadrat is analyzed [using Morueta-Holme 2016 methods, and called the Quadrat correlation matrices (QCM) analysis] as the first assessment for statistical spatial correlation. Standardized effect size (SES) for these correlations (above null distribution and adjusted for multiple comparisons by Bonferroni correction) are calculated. Only pairs of cells displaying SES with FDR $q < 0.05$ are submitted for cross pair correlation function (cross-PCF) analysis which assesses co-location. None of the control ROIs, passed the QCM analysis stage and therefore could not progress to the spatial analysis (cross-PCF) stage. This finding is largely due to the much smaller numbers of immune cells in the healthy control lungs, and also because there are hardly any neutrophils in normal lungs (as is expected since healthy lungs do not have resident neutrophils). This is confirmed by formal abundance measurements in the HC lungs – see Figure 3B above) but to demonstrate this visually, we further stained 4 HC lung sections with CD15 (neutrophil) and CD8 (CD8 T cells) antibody by immunofluorescence (and 4 COVID lung sections), and affirmed that there were hardly any CD8 T cells or neutrophils for the same amount of tissue in HC lung sections, in contrast to COVID lungs (Figure 4).

Figure 4. Immunofluorescence of CD15 expressing cells (neutrophils)(red) and CD8 T cells (green) in COVID lungs (top panel) and healthy lung sections. There were very few neutrophils and CD8 T cells in healthy lung sections in contrast to COVID samples. Scale bar – 10 um.

We did consider staining the additional HC lung sections with the IMC panel to further prove this point. However, we calculated that the re-conjugation of metal tagged antibodies, fresh antibodies and acquisition time for 4x 2 mm² ROIs on Hyperion would cost a total of £11,000 and unlikely to add to what immunofluorescence and H&E staining had demonstrated – i.e scarce immune cells, specifically lack of neutrophils and CD8 T cells, which would preclude progression of the sections to spatial analysis in our pipeline as they will not pass the QCM analysis.

[REDACTED]

Figure 2A has been added to the revised paper, together with the Demographic data.

[FIGURE REDACTED]

To understand better the immune cell distribution in the COVID-19 lungs, this study the authors developed some mathematical tools (radial connectivity map, topological correlation map) to search for statistically significant co-locations amongst immune and structural cells identified using 37-plex imaging mass cytometry. They have found a highly active cluster of immature neutrophils and cytotoxic CD8 T cells that was spatially linked with alveolar progenitor cells. They have also found that CD8 T cells, monocytes and immature neutrophils are linked to disease severity. The authors have provided their pipeline and the visual-analytical tool software.

Overall it is an interesting, well-written manuscript that provides new knowledge about the cellular immune landscape in the COVID-19 lung: cell numbers, phenotypes, spatial associations between different immune cell types. Therefore, the manuscript deserves publication.

Thank you very much for the positive remarks. We are glad you found the paper interesting and deserving of publication. Thank you!

Reviewer #3 (Remarks to the Author):

The manuscript entitled “Unbiased single cell spatial analysis localises active immature neutrophil-CD8 T cell clusters to alveolar progenitor cells in fatal COVID-19 lungs” by Weeratunga et al, performed a single cell spatial analysis of the immune-structural cell interactions in COVID-19 lungs. The investigators reveal a detailed cellular map, highlight newly identified clusters and offer a pipeline for future spatial analyses.

Recommendation: Redirect to a methods or resources journal eg Nature Methods

Thank you for your time in reviewing our M/S. Please find a point-by-point response below in bold. Figures are embedded in the appropriate points in this document but provided in higher resolution in the Appendix.

Major/General comments:

1. The strength of this work clearly lays on the quality and type of samples utilised, as well as the pipeline developed and made available. As such, this reviewer believes that the paper would suit another type of journal, with a stronger focus on methodology and technical/scientific resources.

Thank you for this comment. We agree that the quality and type of samples are strengths and the spatial analysis pipeline is a leading platform in the field, utilising mathematical and statistical power to identify spatial connections which otherwise would not be possible from current spatial methods and traditional histopathology analysis. However, combined with the design of the IMC multiplex panel, it goes beyond this to show a very important immunological message – that **immature** neutrophils are found in lung tissue of fatal COVID pneumonitis, clustered with CD8 T cells to form immunologically active entities which spatially and temporally associates with injured type II alveolar epithelium, and the point of maximal alveolar damage. The presence of immature neutrophils has been observed the blood of patients with severe COVID and sepsis but its presence and consequence in tissue is unknown. This is a major advance in immunological understanding of COVID and potentially also other severe and injurious viral infection. The development of the method and biological interpretation was made possible by bringing together scientists, mathematicians, biologists, immunologists, clinicians with specialism in lung histopathology and lung disease, and clinician scientists. We felt Nature Communication was a particularly suitable platform due to its appeal to all ‘tribes’ of scientists. The message, beyond that of portraying landscape was made possible by the mathematics, and demonstrates the power of collaborative science between many types of scientists.

2. Indeed, the paper is purely descriptive and the interpretation of any interesting findings such as the presence of “neutrophil-CD8” and “monocyte-endothelial” clusters is at least speculative

Thank you for this comment. You are right that the paper is descriptive, though we hope that it is an important finding. We are keen to highlight the following:

Figure showing completely destroyed lung structure in COVID (B) and massive cellular infiltrate compared to normal lung (A).

The finding of this pairwise connection would not be possible without the mathematical algorithms that we have developed. Identification of a pair of cells that is consistently and statistically significantly connecting with each other in this space is akin to finding a needle in a haystack as tissue in COVID patients (and other similarly, severely damaged lung disease) is completely destroyed, bears no resemblance to usual tissue and filled with a massive amount of immune infiltrate (see Figure on left).

To discover any spatial association beyond random position is therefore quite notable and is only achievable with accurate annotation and unbiased mathematical methods. These pairings became obvious when we went back to examine the immature neutrophil-CD8 T cell staining by immunofluorescence (Figure 2G in manuscript).

Another thing to note is that the colocalization is not purely on physical basis but also accompanied by increase in IFN γ , type I IFN and granzyme B protein expression which was only observed in CD8 and immature neutrophils that were found together.

We have added your point in the manuscript and ensured that we have not over-done the interpretation.

3. Additional spatial single cell omics data from the same specimens could confirm specific interesting findings of this work. For example, the discrepancy between transcriptomic and celomic approaches in regards to conclusions on the role of neutrophils in disease due to their low RNA content.

Thank you.

Our key finding is the presence of CD8 T cell- immature neutrophil foci with high immunological activity which were found spatially and temporally with the stage of disease with maximal alveolar injury - diffuse alveolar damaged states or DAD. Linked to these was a network of monocytes and megakaryocyte, forming nidus of inflammation.

We understand that we are asked to supplement our protein-led spatial finding with two further information from single cell spatial transcriptomic data:

- i. confirmation that the aforementioned finding is also observed in single cell spatial transcriptomic data and
- ii. support for the discrepancy between transcriptomic and celomic approaches in regards to conclusions on the role of neutrophils in disease due to their low RNA content.

We have taken the approach of addressing point (ii) first, as presence of a neutrophil transcriptomic signature is required to annotate neutrophils and is a prerequisite to seeking out the immature neutrophil-CD8 immune active foci, and thenceforth, the monocyte-megakaryocyte -endothelium-neutrophil-CD8 nidus. We reasoned that without the ability to identify neutrophils, we would not be able to use single cell spatial transcriptomic studies to confirm/support/expand our key findings.

To put the work in context, the methods and analyses for spatial single cell omics technology in tissue are still very much in its infancy, and FFPE-compatible protocols have only just been established and available commercially this year. FFPE processing and storage was a safety necessity for harvesting of COVID lungs. To date, there are no published true single cell resolution spatial transcriptomic data in COVID lungs, though there is one paper using Visium[®] (from 10X platform) (Mothes et al., 2023) which extracts RNA from small defined spots on tissue comprising approximately 50 cells (hence not truly 'single' cell). Currently there are three technologies for single spatial cell transcriptomics available for FFPE tissues – CosMx (Nanostring), MERSCOPE[®] and Xenium (10X)[®], of which MERSCOPE and 10X Xenium are early access commercial technologies (i.e. limited availabilities). There has only been one paper published using CosMx technology on lung cancer (He et al., 2022), one on mouse brain using MERSCOPE (Emanuel and He, 2021) and none on 10X Xenium. We have just acquired the MERSCOPE hardware and amongst the first in the field to work on the staining protocol for FFPE samples (which is what our remaining lung sections are stored as) for MERSCOPE. This has still not been through the final optimisations. The bioinformatic and mathematical analysis to integrate single cell and spatial

location in tissue in the lungs, (a challenging organ compared to solid organs) will be quite a feat, and likely to be a paper in itself.

However, we have established GeoMx® (Nanostring) analysis in FFPE stored lung samples (Cross et al., 2023). GeoMx extracts RNA for small areas of interest (AOI), typically comprising about 500-1000 cells in tissue.

For this revision, we performed further analyses and studies as follows:

We extracted the RNA sequence data from AOIs (n=46) in three COVID lung samples that were used in our paper, and matched these samples to how we have labelled them in our paper [defined by the dominant histopathology in the samples - alveolitis (ALV), diffuse alveolar damage (DAD) and organising pneumonia (OP)]. We then performed two analyses – firstly we compiled the differential expressed gene list between the three states (using DESeq2) and performed a pathway analysis using Reactome(Griss et al., 2020) (Figure 6). Here, we found upregulation of genes associated with neutrophil activation when comparing DAD to OP and ALV, matching our findings by IMC in the manuscript. In particular, S100A8 (highly expressed in neutrophils and a feature of degranulation) and CXCL10 (chemokine related to neutrophils trafficking) were highly upregulated, supporting trafficking of neutrophil to the tissue at the DAD phase(Ichikawa et al., 2013). High expression of CXCL9 a key chemokine in T cell extravasation into tissue supports finding of T cells (e.g. CD8 T cells) in these AOIs. We have put these data in the revised manuscript (Figure S11).

Figure 6. Analysis of gene expression profile of specific areas from three patients with COVID. Gene expression profile from representative AOIs (n=46 AOIs from 3 COVID lung sections) (published in Cross A et al 2022) was derived using a probe hybridization panel for 1852 genes (Nanostring GeoMx) (Cross et al 2022). These were downloaded, re-organised according to ALV, DAD and OP histopathology states and re-analysed. DEGs derived using Enhanced Volcano package showed that the gene expression profile in DAD is enriched with neutrophil-related genes (CXCL10) when compared to ALV and OP. Pathway analysis with Reactome showed interleukin and tyrosine kinase signaling pathways as those with top gene ratios in DAD vs ALV (C) and cytokine signaling and neutrophil degranulation in DAD v OP (D). Both these data support excess neutrophil-related activity in ALV compared to the other histopathology states, enhancing our current data but cannot conclude on presence and location of neutrophils.

In the next step, we ask if we can deconvolute the gene expression profile (for 988 cells per AOI) into single neutrophils data in order to match these to ‘celomic’ neutrophilic presence. To do this, we selected an AOI each from the three samples, and ablate the matched AOI on the next slide

(representing the consecutive lung) which was stained with a 30-plex metal-tagged IMC antibody panel to determine 12 key cells (Table 2). 1.27mm² were ablated in total. Bulk transcriptomic profile for a mean of 988 cells per AOI was then deconvoluted with Spatial DeconR (Danaher et al., 2022) (Figure 7) using the cell profile matrix, lung_plus_neutrophil (Desai et al., 2020). This cell profile matrix was constructed from the Human Lung Cell Atlas reference data (Travaglini et al., 2020) (healthy tissue from lung cancer patients undergoing lobectomy and their peripheral blood) appended with an additional neutrophil profile from non-small cell lung cancer tissue (Zilionis et al., 2019). This reference dataset is crucial and selected as it contained neutrophils in human lungs, necessary for annotation of our clusters as we are specifically looking for neutrophils.

We found presence of celomic neutrophils in all samples by imaging mass cytometry (IMC) (Figure 7A-B) but none by transcriptomic deconvolution (Figure 7C), addressing point (ii) raised by the reviewer – that spatial IMC and single cell deconvolution showed a discrepancy between transcriptomic and celomic approach in identification of neutrophils. Thus, single cell deconvolution of transcriptomic data from a defined number of cells matched to the same area of tissue failed to identify neutrophils while IMC did.

Figure 7. Comparing neutrophil identification by celomic and transcriptomic approaches using Nanostring GeoMx acquired data Three lung sections (Cun 4,5 and 6) were used. In each patient, two consecutive lung sections were obtained, location-matched areas of interest (AOIs) were ablated, and one subjected to multiplex IMC staining to identify neutrophils, macrophage and CD8 T cells; and the second for RNA analysis (Nanostring GeoMx). In (A), circle shows AOI with macrophages (CD68+ cells in green), CD 8 T cells (red) and CD15+ neutrophils in white (see Table 2 for overall phenotypic markers for each cell type). Left panel shows all three and right panel only the neutrophils in the same AOI. B. Formal abundance analysis showing numerical presence of neutrophils in Cun 4 and 6. C. Heat map of abundance of cell types deconvoluted from transcriptomic data from the same AOI as (A). No neutrophils were detected; despite clear detection of these cells by IMC in Cun 4 and 6 observed on imaging (A) and numerical analyses (B).

To validate this finding further, we performed the same analysis but on a publicly deposited dataset - from the only other single cell deconvoluted transcriptomic study on COVID lungs (Mothes et al., 2023) (n=12 severe COVID compared to n=3 non-COVID pneumonia). This study used Visium which extracts RNA from 50 cells within a spot and then provide the entire transcriptome for single cell deconvolution (thus better resolution than Nanostring GeoMx which extracts RNA from about 500 cells and provides transcripts for 500-2000 genes, according to selected nucleic acid probes, for deconvolution). We reasoned that this would provide a further test for celomic vs single cell transcriptomic identification of neutrophils, albeit on different COVID patients to ours.

As Visium provides gene expression profile for the entire transcriptome for each of its spot cluster, we were able to interrogate presence of neutrophil signature in the spots with an enhanced neutrophil gene set. To derive this gene set, we used the neutrophil reference dataset used for our Nanostring GeoMx analysis (the Zilionis et al dataset from non-small cell lung cancer tissue) and obtained the top 100 differential expressed genes (DEGs) for neutrophils compared to other immune cells (selected on $\text{avg_log2_fold_change} > 0.5$ and adjusted p value < 0.05). 18 of these DEGs were found expressed with relative specificity in neutrophils and used as the 'neutrophil module' to interrogate each of the Visium cluster spots for presence of neutrophils. A neutrophil module score for each of the Visium spots (representing expression of the neutrophil signature) was generated using the *addmodulescore* function in Seurat (<https://satijalab.org/seurat/reference/addmodulescore>) (Hao et al., 2021; Tirosh et al., 2016) (Figure 8A-C) and compared to protein (or 'celomic') analysis for neutrophil using multiplex immunofluorescence data. The latter was created from the multiplex immunofluorescence images and data deposited by Mothes et al (location provided in Figure 1 in Mothes et al.) (Mothes et al., 2023). From Mothes' data, we generated merged images for the immunofluorescence channels; DAPI, Col – IV, CD66b, CD3 and CD14 (Figure 8D-F) showing neutrophils, CD3 T cells, monocytes and structural outline in the matched area.

Even with this targeted approach, we did not observe transcriptomically-deconvoluted neutrophils despite clear presence of celomic neutrophils determined by protein staining in matched sections of lung in this study (Figure 8D-F).

These findings strongly support the deconvolution results using GeoMx in our lung COVID tissue and the proposal by Reviewer 3 that there is a discrepancy between transcriptomic and celomic approaches in analyzing tissue data, necessitating the use of protein or cell identification as we have done in our paper. The inability to identify the neutrophilic counterpart in the RNA profile also meant that we are unable to explore the transcriptomic composition of the neutrophils as identified by IMC as we require a spatially sensitive information of the cells (i.e. we need to identify those neutrophils that are co-located with CD8 T cells) to examine the relevant transcriptome.

We hope this extended piece of work provide a powerful support for the need for protein/'celomic' studies to complement transcriptomic work as raised by this reviewer. We have noted the importance of this comparison and added these to the paper and believe the paper has been greatly strengthened by addressing this comment.

References

- Cross, A.R., de Andrea, C.E., Villalba-Esparza, M., Landecho, M.F., Cerundolo, L., Weeratunga, P., Etherington, R.E., Denney, L., Ogg, G., Ho, L.-P., *et al.* (2023). Spatial transcriptomic characterization of COVID-19 pneumonitis identifies immune circuits related to tissue injury. *JCI Insight* **8**.
- Danaher, P., Kim, Y., Nelson, B., Griswold, M., Yang, Z., Piazza, E., and Beechem, J.M. (2022). Advances in mixed cell deconvolution enable quantification of cell types in spatial transcriptomic data. *Nature Communications* **13**, 385.
- Desai, N., Neyaz, A., Szabolcs, A., Shih, A.R., Chen, J.H., Thapar, V., Nieman, L.T., Solovyov, A., Mehta, A., Lieb, D.J., *et al.* (2020). Temporal and spatial heterogeneity of host response to SARS-CoV-2 pulmonary infection. *Nature Communications* **11**, 6319.
- Emanuel, G., and He, J. (2021). Using MERSCOPE to Generate a Cell Atlas of the Mouse Brain that Includes Lowly Expressed Genes. *Microscopy Today* **29**, 16-19.
- Griss, J., Viteri, G., Sidiropoulos, K., Nguyen, V., Fabregat, A., and Hermjakob, H. (2020). ReactomeGSA - Efficient Multi-Omics Comparative Pathway Analysis. *Molecular & cellular proteomics* : *MCP* **19**, 2115-2125.
- Hao, Y., Hao, S., Andersen-Nissen, E., Mauck, W.M., III, Zheng, S., Butler, A., Lee, M.J., Wilk, A.J., Darby, C., Zager, M., *et al.* (2021). Integrated analysis of multimodal single-cell data. *Cell* **184**, 3573-3587.e3529.
- He, S., Bhatt, R., Brown, C., Brown, E.A., Buhr, D.L., Chantranuvatana, K., Danaher, P., Dunaway, D., Garrison, R.G., Geiss, G., *et al.* (2022). High-plex imaging of RNA and proteins at subcellular resolution in fixed tissue by spatial molecular imaging. *Nature Biotechnology* **40**, 1794-1806.
- Ichikawa, A., Kuba, K., Morita, M., Chida, S., Tezuka, H., Hara, H., Sasaki, T., Ohteki, T., Ranieri, V.M., dos Santos, C.C., *et al.* (2013). CXCL10-CXCR3 enhances the development of neutrophil-mediated fulminant lung injury of viral and nonviral origin. *American journal of respiratory and critical care medicine* **187**, 65-77.
- Mothes, R., Pascual-Reguant, A., Koehler, R., Liebeskind, J., Liebheit, A., Bauherr, S., Philipsen, L., Dittmayer, C., Laue, M., von Manitius, R., *et al.* (2023). Distinct tissue niches direct lung immunopathology via CCL18 and CCL21 in severe COVID-19. *Nature Communications* **14**, 791.
- Tirosh, I., Izar, B., Prakadan, S.M., Wadsworth, M.H., 2nd, Treacy, D., Trombetta, J.J., Rotem, A., Rodman, C., Lian, C., Murphy, G., *et al.* (2016). Dissecting the multicellular ecosystem of metastatic melanoma by single-cell RNA-seq. *Science (New York, NY)* **352**, 189-196.
- Travaglini, K.J., Nabhan, A.N., Penland, L., Sinha, R., Gillich, A., Sit, R.V., Chang, S., Conley, S.D., Mori, Y., Seita, J., *et al.* (2020). A molecular cell atlas of the human lung from single-cell RNA sequencing. *Nature* **587**, 619-625.
- Zilionis, R., Engblom, C., Pfirschke, C., Savova, V., Zemmour, D., Saatcioglu, H.D., Krishnan, I., Maroni, G., Meyerovitz, C.V., Kerwin, C.M., *et al.* (2019). Single-Cell Transcriptomics of Human and Mouse Lung Cancers Reveals Conserved Myeloid Populations across Individuals and Species. *Immunity* **50**, 1317-1334.e1310.

REVIEWERS' COMMENTS

Reviewer #1 (Remarks to the Author):

I have re-reviewed the paper by Ho et al submitted for publication to Nature Communications. The revised paper has increased my enthusiasm for this work. The addition of the tissue selection criteria to the paper has improved the impact and I am glad to see that addition. The authors make an argument in their rebuttal about the cost of the method for larger studies to increase the power of the study. While I understand the issue it is also an affirmation of the challenges of the method. The cell counts by themselves are usefull and add value, but the patient number remains limited. The second study mentioned in confidence provides a broader perspective, but as this is not referenced in the paper, the readers are unaware of the linked findings. I would suggest to include the reference as suggested as a minimum. Other than this remaining issue, my initial concerns have been well addressed by the authors.

Reviewer #3 (Remarks to the Author):

The reviewers comments have been now adressed and there are no further considerations.

Point-by-point response to reviewers' comments (in green and bold):

REVIEWERS' COMMENTS

Reviewer #1 (Remarks to the Author):

I have re-reviewed the paper by Ho et al submitted for publication to Nature Communications. The revised paper has increased my enthusiasm for this work. The addition of the tissue selection criteria to the paper has improved the impact and I am glad to see that addition. The authors make an argument in their rebuttal about the cost of the method for larger studies to increase the power of the study. While I understand the issue it is also an affirmation of the challenges of the method. The cell counts by themselves are useful and add value, but the patient number remains limited. The second study mentioned in confidence provides a broader perspective, but as this is not referenced in the paper, the readers are unaware of the linked findings. I would suggest to include the reference as suggested as a minimum. Other than this remaining issue, my initial concerns have been well addressed by the authors.

Thank you for your comment. We will indeed include the reference in the second study which is nearly ready for submission.

Reviewer #3 (Remarks to the Author):

The reviewers comments have been now addressed and there are no further considerations.
Thank you.